# Generating active metal/oxide reverse interfaces through coordinated migration of single atoms

Lina Zhang[1,2,9], Shaolong Wan[1,9], Congcong Du[1,9], Qiang Wan[3,9], Hien Pham[4], Jiafei Zhao[1], Xingyu Ding[1], Diye Wei[1], Wei Zhao[5], Jiwei Li[1], Yanping Zheng[1], Hui Xie[1], Hua Zhang[1], Mingshu Chen[1], Kelvin H. L. Zhang[1], Shuai Wang[1,2], Jingdong Lin[1], Jianyu Huang[6], Sen Lin[3]✉, Yong Wang[7], Abhaya K. Datye[4], Ye Wang[1,2]✉ & Haifeng Xiong[1,2,8]✉

Identification of active sites in catalytic materials is important and helps establish approaches to the precise design of catalysts for achieving high reactivity. Generally, active sites of conventional heterogeneous catalysts can be single atom, nanoparticle or a metal/oxide interface. Herein, we report that metal/oxide reverse interfaces can also be active sites which are created from the coordinated migration of metal and oxide atoms. As an example, a $Pd_1/CeO_2$ single-atom catalyst prepared via atom trapping, which is otherwise inactive at 30 °C, is able to completely oxidize formaldehyde after steam treatment. The enhanced reactivity is due to the formation of a $Ce_2O_3$-Pd nanoparticle domain interface, which is generated by the migration of both Ce and Pd atoms on the atom-trapped $Pd_1/CeO_2$ catalyst during steam treatment. We show that the generation of metal oxide-metal interfaces can be achieved in other heterogeneous catalysts due to the coordinated mobility of metal and oxide atoms, demonstrating the formation of a new active interface when using metal single-atom material as catalyst precursor.

Recent changes in weather, ecosystems and more show that the earth's climate is changing due to carbon-based gas emission, such as greenhouse gas[1] and volatile organic compounds (VOCs). VOCs emitted by biogenic and anthropogenic sources, e.g., O- and N-containing hydrocarbons, are detrimental to human health and must be eliminated. According to a 2019 report of the World Health Organization (WHO), 99% of the world's population lives with an atmospheric quality that does not meet the WHO guidelines, and outdoor air pollution was estimated to have caused 4.2 million premature deaths in 2016[2]. Heterogeneous catalysts can play a major role in abating emissions of VOCs to the atmosphere.

Catalytic oxidation is a promising approach for VOC abatement due to its high efficiency and energy saving. Supported noble metal catalysts are preferred because of their high specific activity and

[1]State Key Laboratory of Physical Chemistry of Solid Surfaces, Collaborative Innovation Center of Chemistry for Energy Materials, College of Chemistry & Chemical Engineering, Xiamen University, Xiamen 361005, China. [2]Innovation Laboratory for Sciences and Technologies of Energy Materials of Fujian Province, Xiamen 361102, China. [3]State Key Laboratory of Photocatalysis on Energy and Environment, College of Chemistry, Fuzhou University, Fuzhou 350100, China. [4]Department of Chemical and Biological Engineering and Center for Micro-Engineered Materials, University of New Mexico, Albuquerque, NM 87131, USA. [5]Institute for Advanced Study, Shenzhen University, Shenzhen 518060, China. [6]Clean Nano Energy Center, State Key Laboratory of Metastable Materials Science and Technology, Yanshan University, Qinhuangdao 066000, China. [7]Voiland School of Chemical Engineering and Bioengineering, Washington State University, Pullman, WA 99164, USA. [8]Fujian Key Laboratory of Rare-earth Functional Materials, Fujian Shanhai Collaborative Innovation Center of Rare-earth Functional Materials, Longyan 366300, China. [9]These authors contributed equally: Lina Zhang, Shaolong Wan, Congcong Du, Qiang Wan. ✉e-mail: slin@fzu.edu.cn; wangye@xmu.edu.cn; haifengxiong@xmu.edu.cn

resistance to deactivation. However, high cost and limited supply of noble metals require significantly improved atom efficiency when designing the noble metal-based catalysts. Metal single-atom catalysts (SACs) provide the highest atom efficiency due to the dispersion of the metal and have the potential to efficiently catalyze the oxidation of VOCs. However, metal SACs are not always as active as their nanocatalyst counterparts[3], hence strategies must be developed to improve the catalytic performance of metal SACs. Such strategies include the treatments of reduction, or oxidation-reduction-oxidation (O-R-O)[4] and also steam-treatment[5]. For example, a thermally-stable $Pt_1/CeO_2$ SAC prepared by atom-trapping was not active at low temperatures in CO oxidation. After treating the $Pt_1/CeO_2$ SAC in reducing atmospheres[6,7], the catalytic activity of the thermally-stable $Pt_1/CeO_2$ SAC in CO oxidation could be improved. Alternatively, the $Pt_1/CeO_2$ SAC demonstrated improved reactivity in CO oxidation after treating in steam at 750 °C, which modified the coordination structure of the Pt species, while maintaining the atomic dispersion of the Pt[8].

To achieve high catalyst reactivity and stability in heterogeneous catalysis, the active sites of supported metal catalysts must be carefully designed and constructed[9–13]. Depending on the catalytic reaction of interest, the active sites of conventional heterogeneous catalysts may involve metal single atoms[14] or metal nanoparticles[15]. Here, we report the generation of an active interface in heterogeneous catalysis, oxide-metal domain interfaces, which can only be achieved from the coordination migration of both oxide and metal single atoms on atom-trapped single-atom catalyst. The new oxide-metal domain interfaces were generated by steam-treatment at 750 °C of an inactive $Pd_1/CeO_2$ single-atom catalyst ($Pd_1/CeO_2$-AT) prepared via atom trapping (800 °C in air). The obtained material ($Pd/CeO_2$-AT-S) generates $Ce_2O_3$-Pd nanoparticle interfaces, which can efficiently oxidize formaldehyde (HCHO) at room temperature.

## Results and discussion
### Catalytic performance for HCHO oxidation
The catalytic performance of the $Pd_1/CeO_2$-AT and $Pd/CeO_2$-AT-S catalysts was tested in HCHO oxidation by ramping the temperature from −20 °C to 200 °C (Fig. 1a). As can be seen, the atom-trapped Pd single-atom catalyst ($Pd_1/CeO_2$-AT) reached 100% HCHO conversion at 180 °C, as compared to the $Pd/CeO_2$-AT-S catalyst which showed complete HCHO conversion at ~30 °C (Fig. 1a). Even at 0 °C, the catalyst was still sufficiently active in achieving 50% conversion. The reaction rate (noted as $r$ value) of $Pd/CeO_2$-AT-S was 100 $\mu mol \cdot g_{Pd}^{-1} \cdot s^{-1}$ at 30 °C, which is better than the Pd-based catalysts reported in the literature (Supplementary Table 1). Moreover, at HCHO conversion of ~92%, the $Pd/CeO_2$-AT-S catalyst is very stable and no deactivation is observed over a 7 h time-on-stream run (Fig. 1b). This indicates that the steam-treated $Pd/CeO_2$-AT-S catalyst is an excellent low-temperature HCHO oxidation catalyst. It is important to study the effect of relative humidity (RH) on the catalytic oxidation of HCHO at room temperature. Under 10% relative humidity (Supplementary Fig. 1), the reactivity of $Pd/CeO_2$-AT-S catalyst remained stable. It indicates that the $Pd/CeO_2$-AT-S catalyst has good tolerance under humidity conditions. We also examined the effect of steam treatment on the catalytic performance of a reference $Pd_1/CeO_2$ single-atom catalyst prepared by conventional impregnation ($Pd_1/CeO_2$-I) in the reaction (Supplementary Fig. 2). The $Pd_1/CeO_2$-I sample shows a similar catalytic reactivity and apparent activation energy of ~82.4 kJ·mol⁻¹ as the atom-trapped $Pd_1/CeO_2$-AT catalyst (Fig. 1a, c), indicating a similar reaction mechanism and active site for the two Pd single-atom catalysts. However, the steam-treated $Pd_1/CeO_2$-I catalyst ($Pd/CeO_2$-I-S) shows very low activity (<10%) at 30 °C in HCHO oxidation, as opposed to the 100% HCHO conversion for the $Pd/CeO_2$-AT-S catalyst. Moreover, the $Pd/CeO_2$-I-S catalyst deactivates rapidly at 30 °C (Fig. 1b). Kinetic measurements reveal that the reference $Pd/CeO_2$-I-S catalyst shows an apparent activation energy of ~34.4 kJ mol⁻¹, which is significantly

higher than that on the $Pd/CeO_2$-AT-S catalyst (~15.5 kJ mol⁻¹) (Fig. 1c). Therefore, the atom-trapped $Pd_1/CeO_2$-AT single-atom catalyst is unique because the steam-generated $Pd/CeO_2$-AT-S catalyst presents exceptional low-temperature catalytic reactivity in HCHO oxidation.

### Structure characterization
To understand the reason for the enhanced reactivity of the $Pd/CeO_2$-AT-S catalyst, we used several characterization techniques to investigate the catalyst structure (Supplementary Table 3). XRD pattern of the $Pd_1/CeO_2$ SAC prepared by atom-trapping ($Pd_1/CeO_2$-AT) only shows the diffraction peaks of $CeO_2$ support because the Pd species are atomically dispersed (Supplementary Fig. 2 and Supplementary Fig. 3). In contrast, the XRD pattern of the $Pd/CeO_2$-AT-S shows the presence of a small diffraction peak at 2θ of 40.1°, corresponding to the metal Pd (111) (Supplementary Fig. 4). Therefore, the steam treatment of the $Pd_1/CeO_2$-AT catalyst changed the atomic dispersion of Pd species on $Pd_1/CeO_2$-AT and caused the formation of metal Pd particles. The presence of Pd nanoparticle on the $Pd/CeO_2$-AT-S was also confirmed by the EXAFS showing Pd-Pd scattering (Supplementary Fig. 6). *XPS* analysis of the $Pd_1/CeO_2$-AT shows a strong Pd 3$d$ signal corresponding to $Pd^{2+}$ species, while the Pd 3$d$ *XPS* spectrum on the $Pd/CeO_2$-AT-S is barely seen and only shows the weak signal corresponding to the metal $Pd^0$ (Fig. 1d). This possibly suggests that the Pd single atoms in the $Pd_1/CeO_2$-AT sample migrated during high-temperature steam treatment to form metal Pd nanoparticles on the $Pd/CeO_2$-AT-S, which was covered by the mobile support, leading to an attenuated signal for $Pd^0$. The covering of the metal Pd nanoparticle by support was also confirmed by the Low Energy Ion Scattering (LEIS), which is exquisitely outmost surface sensitive. The LEIS spectrum of the $Pd/CeO_2$-AT-S catalyst shows the absence of Pd signal on the $Pd/CeO_2$-AT-S catalyst (Fig. 1e), as compared to the significant peak of Pd on the $Pd_1/CeO_2$-AT catalyst. However, multiple scans of LEIS on the $Pd/CeO_2$-AT-S catalyst show that the peak intensity of the Pd signal increases with continuous scanning, indicating the increased surface concentration due to sputtering (Supplementary Fig. 7). The surface species on the $Pd/CeO_2$-AT-S catalyst were determined by CO-DRIFTS. As shown in the Fig. 1f, two weak peaks of CO adsorption at 2164 cm⁻¹ and 1970 cm⁻¹ were observed on the $Pd/CeO_2$-AT-S catalyst, as opposed to the adsorption peaks at 2087 cm⁻¹ and 1970 cm⁻¹ for the $Pd_1/CeO_2$-AT. For the $Pd_1/CeO_2$-AT catalyst, the band at 2087 cm⁻¹ corresponding to the CO on the atop sites of metallic Pd nanoparticle is seen on the Pd SAC because the Pd atoms in the $Pd_1/CeO_2$-AT are mobile and readily transforms from their oxidized state into a reduced state during CO oxidation, which is consistent with previous studies on a $Pd/CeO_2$ SAC[16–18]. For the $Pd/CeO_2$-AT-S catalyst, the adsorption peak at 2164 cm⁻¹ is attributed to the CO adsorption on $Ce^{3+}$ [19–22] and the peak at 1970 cm⁻¹ is attributed to CO on the bridge sites on Pd nanoparticles. In contrast, the CO-DRIFTS of the $Pd/CeO_2$-I-S sample only presents the CO adsorption on Pd nanoparticles (Supplementary Fig. 8). Therefore, the $Pd/CeO_2$-AT-S catalyst prepared from atom-trapped $Pd_1/CeO_2$-AT contains both $Ce^{3+}$ species and metal Pd nanoparticles after the high-temperature steam treatment.

Aberration-corrected STEM was used to investigate the morphology and structure of the $Pd/CeO_2$-AT-S catalyst at the atomic scale. Because Pd and Ce have a similar atomic mass, HAADF (high-angle annular dark-field) STEM images cannot provide enough contrast to visualize atomically dispersed or sub-nanometer Pd dispersed on the $CeO_2$ support. However, we can use STEM-EDS mapping to track the location of the Pd species on the ceria. STEM-EDS mapping image (Fig. 2) shows that an irregular-shaped Pd nanoparticle with an average diameter of ~10 nm located on the ceria support after treating the $Pd_1/CeO_2$-AT sample in the high-temperature steam (Fig. 2a-c). High-resolution HAADF-STEM image shows that lattice fringes corresponding to $Ce_2O_3$ were detected on the Pd nanoparticles (Fig. 2d, e and Supplementary Figs. 9 and 10), indicating that the high-

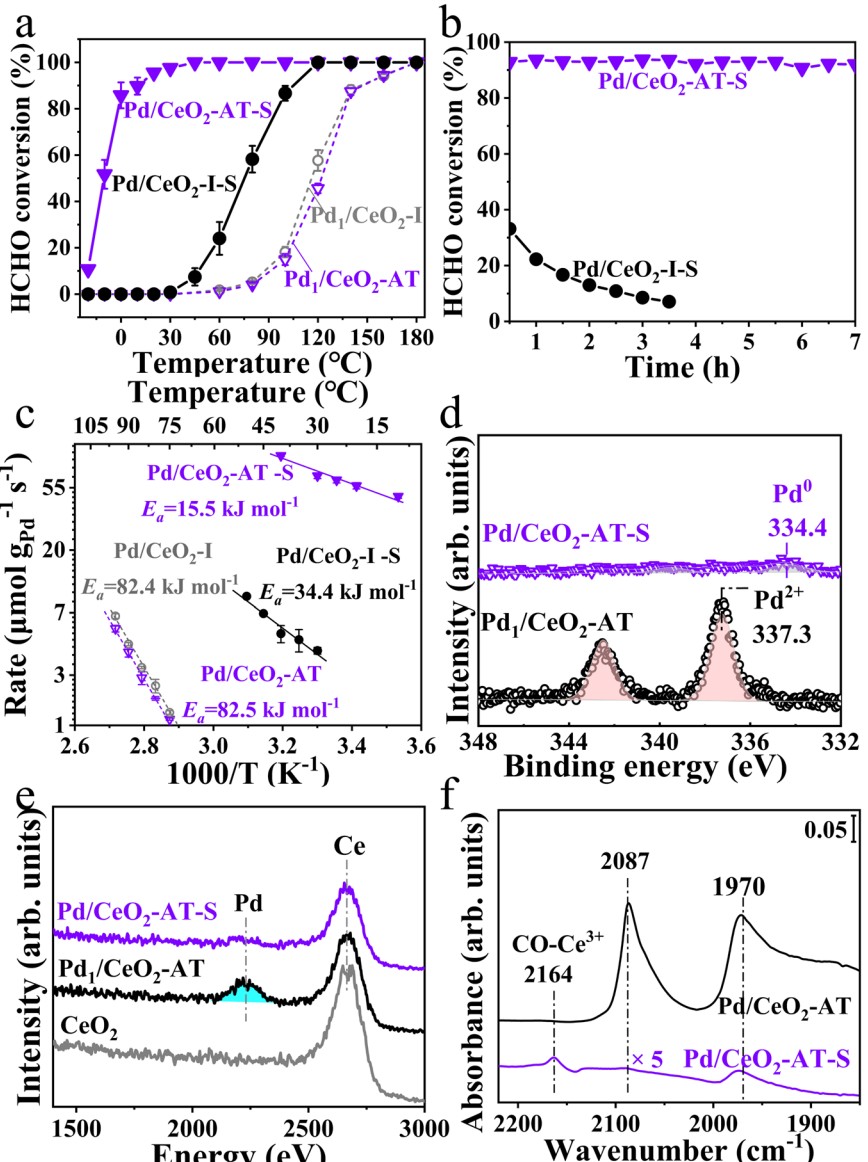

**Fig. 1 | Steam-treated Pd₁/CeO₂-AT SAC prepared by atom-trapping generates a highly active catalyst in formaldehyde oxidation attributed to the formation of Ce³⁺ species. a** HCHO conversion as a function of temperature on catalysts (Pd/CeO₂-AT-S, Pd₁/CeO₂-AT, Pd/CeO₂-I-S, and Pd₁/CeO₂-I). The error bar indicates the standard deviation of the data measured in three tests. Reaction conditions: 400 ppm HCHO, 20 vol% O₂, and N₂ as balance gas, total flow rate: 50 mL·min⁻¹, WHSV (weight hourly space velocity): 60,000 mL·g⁻¹·h⁻¹. **b** HCHO conversion as a function of time on stream on Pd/CeO₂-AT-S and Pd/CeO₂-I-S catalysts at 30 °C. Reaction conditions: 400 ppm HCHO, 20 vol% O₂, and N₂ as balance gas, total flow rate: 50 mL·min⁻¹ and WHSV: 100,000 mL·g⁻¹·h⁻¹. **c** Arrhenius plots of reaction rate

($\mu$mol HCHO$_{converted}$·g$_{Pd}$⁻¹·s⁻¹) over the Pd catalysts supported on CeO₂ before and after steam treatment. Reaction conditions: 520 ppm HCHO, 20 vol% O₂, and balance N₂, total flow rate: 100 mL·min⁻¹. The conversion was kept at <20% by adjusting the catalyst amount. The error bar indicates the standard deviation of the data measured in three tests. **d** *XPS* spectra of Pd *3d* for the Pd₁/CeO₂-AT and Pd/CeO₂-AT-S catalysts. **e** 5 keV ²⁰Ne⁺ HS-LEIS spectra of the CeO₂, Pd₁/CeO₂-AT, and Pd/CeO₂-AT-S catalysts showing the species on the topmost layer of the samples. **f** CO-DRIFTS spectra of the Pd₁/CeO₂-AT and Pd/CeO₂-AT-S catalysts after exposure to a flow of CO/O₂/N₂ for 30 min, followed by degassing in O₂/N₂ at 30 °C for 15 min (arb. units, arbitrary units).

temperature treatment in steam led to the mobility of Pd single atoms to form Pd nanoparticles, simultaneously producing Ce₂O₃ layers on the surface of the Pd metal particle. The existence of Ce₂O₃ is also confirmed by EELS. We performed EELS on the different locations of the STEM image in Fig. 2d. The obtained EELS spectra (Fig. 2f) reveal that the species away from the Pd particle of the Pd/CeO₂-AT-S catalyst are Ce⁴⁺ (region 3)[23], whereas the species associated with the Pd nanoparticle are Ce³⁺ (region 2). Therefore, EELS analysis suggests that Ce³⁺ species are associated with the Pd nanoparticle on the Pd/CeO₂-AT-S catalyst, which agrees with both STEM and CO-DRIFTS results. We therefore conclude that high-temperature steam treatment led to the

formation of Ce₂O₃-Pd domains. A schematic illustration showing the formation of the domain structure is presented in Fig. 2g.

## Reaction active sites of Pd/CeO2-AT-S

We show next that the presence of both Ce₂O₃ and Pd metallic nanoparticles constitutes the active sites of the catalyst in HCHO oxidation by treating the sample in air at progressively higher temperatures (50–250 °C). As seen in Fig. 3a, the Pd/CeO₂-AT-S sample is active after the pretreatment at temperatures of ≤200 °C in air, indicating that the catalyst is stable and against oxidation up to 200 °C. After treating in air at 250 °C, the Pd/CeO₂-AT-S shows no activity for HCHO oxidation

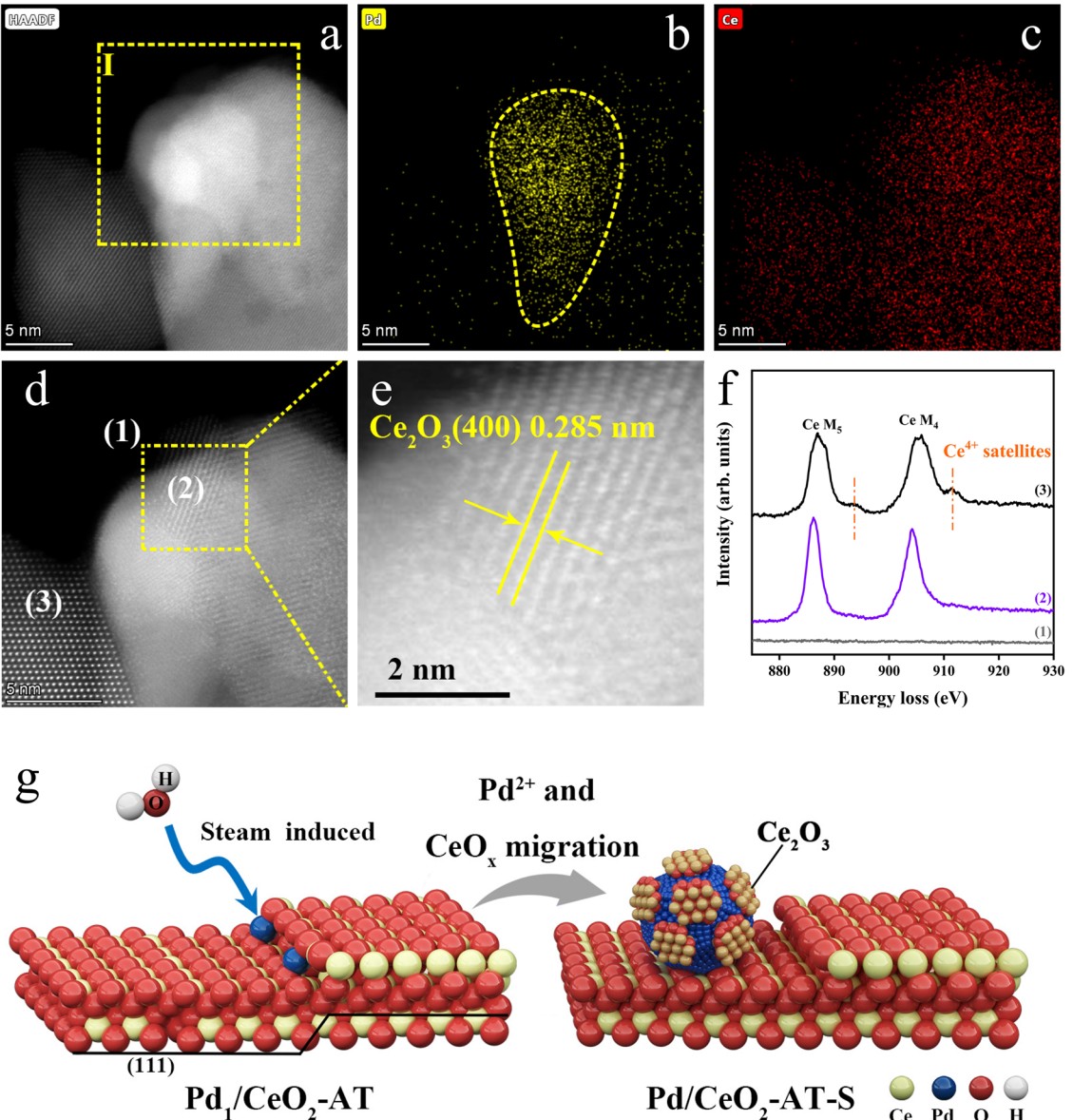

**Fig. 2 | Ce$_2$O$_3$ domains on metallic Pd in the steam-treated Pd$_1$/CeO$_2$-AT SAC.** HAADF-STEM (**a**, **d**, **e**) images and STEM-EDS mapping (**b**, **c**) of the Pd/CeO$_2$-AT-S catalyst. **e** High-resolution lattice fringes of the rectangular region in (**d**, **f**) EELS of the Pd/CeO$_2$-AT-S from different regions are indicated with (1), (2), and (3) in (**d**), which confirm that Pd nanoparticles are covered by the Ce$_2$O$_3$ layer. **g** Schematic illustration showing the structure of the Pd/CeO$_2$-AT-S catalyst derived from steam treatment (750 °C for 9 h, 10 v/v% H$_2$O/Ar) of atom-trapped Pd$_1$/CeO$_2$-AT prepared at 800 °C for 10 h in air.

(Fig. 3a). Figure 3b and Supplementary Fig. 11 present the CO-DRIFTS spectra of the Pd/CeO$_2$-AT-S after pretreatment in air at different temperatures. When the Pd/CeO$_2$-AT-S was pretreated in air at 200 °C, the CO adsorption peaks on both Ce$^{3+}$ and metal Pd are still preserved. After the sample was oxidized at 250 °C in air, the CO peak on metal Pd is seen, while the CO peak on Ce$^{3+}$ (2163 cm$^{-1}$) vanished, indicating the reoxidation of Ce$^{3+}$ in air at 250 °C. The EXAFS results also confirmed that the valence state of Pd did not change (Supplementary Fig. 12) after the Pd/CeO$_2$-AT-S catalyst was oxidized at 250 °C, while the proportion of Ce$^{3+}$ was reduced (Supplementary Figs. 13, 14 and Supplementary Table 4). Combined with the reactivity in Fig. 3a and the fact that metallic Pd nanoparticles only start to be oxidized at temperatures >265 °C[24,25] (Supplementary Fig. 15), the results above suggest that the reactivity of the Pd/CeO$_2$-AT-S is closely related to the presence of Ce$^{3+}$ and the reoxidation of Ce$^{3+}$ at 250 °C leads to the complete deactivation of the Pd/CeO$_2$-AT-S catalyst. The dependence of the reaction rate with O$_2$ partial pressure over the Pd$_1$/CeO$_2$-AT

catalyst showed that the reaction rate has a positive relationship with the HCHO partial pressure and is not related to the O$_2$ partial pressure (Fig. 3c), indicating a Mars-van Krevelen (MvK) reaction mechanism[23] over the Pd$_1$/CeO$_2$-AT SAC. However, kinetic experiments show that the adsorption of O$_2$ on the Pd/CeO$_2$-AT-S catalyst has a typical Langmuir adsorption behavior (Fig. 3c). In contrast, the reaction rate on the catalyst is positive order at low partial pressures of HCHO and then negative order at higher partial pressures of HCHO. This indicates that the sites available for O$_2$ adsorption are occupied by the HCHO at high HCHO partial pressure, leading to a drop in the reaction rate. The adsorption competition toward the sites for O$_2$ and HCHO suggests that a single type of active site is present in the Pd/CeO$_2$-AT-S catalyst. Therefore, the oxidation of HCHO on the Pd/CeO$_2$-AT-S catalyst follows the Langmuir-Hinshelwood (L-H) reaction mechanism of the competing adsorption reaction of the two adsorbed reactants[26]. This Sabatier optimum between reaction rate and reactant partial pressure was often observed in the hydrogenolysis of alkanes[27].

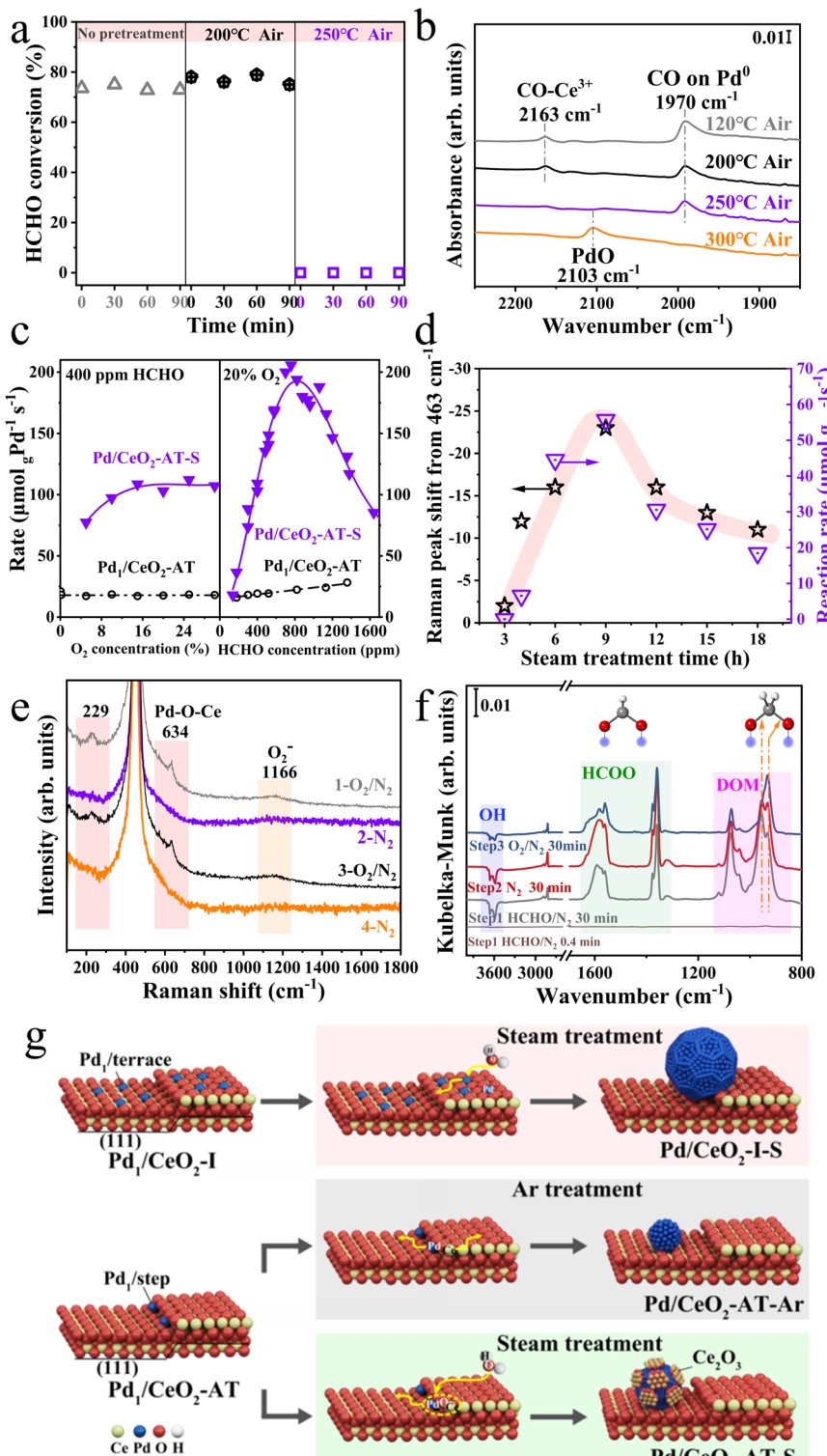

**Fig. 3 | The impact of oxygen vacancies on the reactivity and active sites of the Pd/CeO₂-AT-S catalyst.** **a** HCHO conversion on the Pd/CeO₂-AT-S catalyst treated at different oxidation temperatures. Reaction conditions: 30 °C, 400 ppm HCHO, 20 vol% O₂, and N₂ as balance gas, and total flow rate: 50 mL·min⁻¹. **b** CO-DRIFTS spectra of the Pd/CeO₂-AT-S catalyst treated at different oxidation temperatures after exposure to a flow of CO/O₂/N₂ for 30 min, followed by degassing in a flow of O₂/N₂ for 15 min at 30 °C. **c** The reaction rates on Pd/CeO₂-AT-S and Pd₁/CeO₂-AT catalysts were measured at 30 °C or 120 °C as a function of HCHO or O₂ partial pressure (The effect of internal and external diffusion was eliminated, Supplementary Fig. 16). **d** Reaction rate and Raman peak shift from 463 cm⁻¹ (The $F_{2g}$ peak center of the Pd/CeO₂ catalyst) of the Pd/CeO₂-AT-S catalysts after treating in steam (750 °C, 10 v/v% H₂O/Ar) for different times. **e** In situ Raman spectra of the Pd/CeO₂-AT-S catalyst collected in different atmospheres (O₂/N₂ or N₂) at 120 °C. **f** In situ HCHO-DRIFTS of the Pd/CeO₂-AT-S catalyst after exposure to a flow of HCHO/N₂ for 30 min (curve 2), followed by degassing in N₂ for 30 min (curve 3) and in a flow of O₂/N₂ for 30 min (curve 4) at 30 °C (Inset: the structures of formate and dioxymethylene). **g** Scheme illustrations showing the formation of Pd nanoparticles and Ce₂O₃-Pd nanoparticle domain interface under the treatments of both Ar and steam at 750 °C. The high-temperature steam treatment on the atom-trapped Pd₁/CeO₂-AT resulted in the mobility of Pd-O-Ce ensemble, leading to the generation of Ce₂O₃-Pd nanoparticle domain interface.

We further investigated the effect of steam treatment time (3–18 h) on the structure and catalytic performance of the Pd/CeO$_2$-AT-S catalyst. *XPS* spectra show a stepwise disappearance of the Pd$^{2+}$ peak with increasing the treatment time from 3–18 h (Supplementary Fig. 17), accompanied by the appearance of the Pd$^0$ signal first and then disappearance with the increase of treatment time. Since the high-temperature steam treatment induces the formation of Pd nanoparticles (Supplementary Figs. 6 and 18), these results suggest that the Pd nanoparticle is gradually covered by a Ce$_2$O$_3$ layer upon increasing steam treatment time (Supplementary Fig. 19), agreeing well with the STEM observations. However, the Ce$_2$O$_3$ coverage of Pd nanoparticles beyond a threshold led to a decrease in the catalytic activity. As shown in Supplementary Fig. 14, long-time treatment (>9 h) led to the disappearance of the CO peak on metal Pd nanoparticles due to the complete coverage of Pd nanoparticles, which caused a decreased rate (Supplementary Fig. 20), even though the CO peak on Ce$_2$O$_3$ was still observable. Raman spectroscopy of these treated catalysts shows that the peak assigned to Ce-O-Ce (440–470 cm$^{-1}$)[28] shifts to lower wavenumbers as the treatment time increased from 3 to 9 h (Supplementary Fig. 21). Further increase of the treatment time from 9 to 18 h leads to the upshift of the wavenumber. The shifting of the wavenumber is attributed to the existence of the oxygen vacancies produced by steam treatment on the catalyst[28]. The upshift of the Ce-O-Ce after 9 h treatment is attributed to the decreased surface area of the catalyst and the growth of Pd nanoparticles due to low thermal stability of ceria. The reactivity of these catalysts for formaldehyde oxidation at room temperature shows a volcano relationship with respect to the treatment time. The Pd$_1$/CeO$_2$-AT treated for 9 h is the most active catalyst (Supplementary Fig. 20). The Raman shift from 463 cm$^{-1}$ as a function of treatment time was also determined as shown in Fig. 3d. There is an excellent correlation between the reaction rate and Raman peak shift as a function of treatment time. These results indicate that the reaction rate is closely related to the lattice oxygen vacancies (i.e., the presence of Ce$^{3+}$ species) on the Pd/CeO$_2$-AT-S catalyst, demonstrating a close relationship between Ce$_2$O$_3$ and reaction rate. Since Ce$_2$O$_3$ is always associated with Pd nanoparticles, it suggests that the Ce$_2$O$_3$-Pd domain interface may contribute to the activity because neither Ce$_2$O$_3$ nor Pd nanoparticles itself is the most active site in the HCHO oxidation.

## In situ analysis of reaction mechanisms

We next used in situ Raman spectroscopy and in situ DRIFTS to investigate reaction intermediates over the Pd/CeO$_2$-AT-S and to identify the active sites. The exceptional ability of the Pd/CeO$_2$-AT-S catalyst for activating O$_2$ is confirmed by in situ Raman spectroscopy. We switched between N$_2$ and O$_2$ while recording Raman spectra of the Pd/CeO$_2$-AT-S catalyst at 120 °C, as shown in Fig. 3e. A peak at 448 cm$^{-1}$ was assigned to the Ce-O-Ce vibration which was observed in flowing N$_2$. By switching to flowing O$_2$ (curves 1 and 3), we observed a remarkable increase in the intensity of the Ce-O-Ce vibrations (Supplementary Fig. 22). This indicated the generation of Ce-O species on the sample in the presence of O$_2$ at 120 °C. Furthermore, a band at 634 cm$^{-1}$ corresponding to Pd-O-Ce[29] and at 1166 cm$^{-1}$ corresponding to Ce-O-Ce[30] can be clearly seen with the introduction of O$_2$ flow after switching from a flow of N$_2$. The excellent O$_2$ activation of Pd/CeO$_2$-AT-S was also observed by switching HCHO/N$_2$ and O$_2$/N$_2$ at 25 °C by in situ Raman spectroscopy (Supplementary Fig. 23). However, gas phase oxygen molecules did not directly exchange with the lattice oxygen in the reaction of the formaldehyde oxidation over the Pd/CeO$_2$-AT-S catalyst (Supplementary Fig. 24), corroborating a Langmuir-Hinshelwood (L-H) reaction mechanism. In the oxygen-exchange experiment, the Pd$_1$/CeO$_2$-AT SAC was first treated in H$_2$$^{18}$O (750 °C, 9 h). The obtained Pd/CeO$_2$-AT-S (H$_2$$^{18}$O) catalyst was kept in the flowing HCHO/He (10 mL·min$^{-1}$) at 30 °C. Then, $^{16}$O$_2$ was pulsed into the flow and the products were monitored by an online mass spectrometer, revealing that there are no $^{18}$O species presented in

the product. In addition, we found that Pd/CeO$_2$-AT-S catalyst had stronger oxygen activation capacity and adsorption strength through O$_2$-TPD (Supplementary Fig. 25) and HCHO-TPSR experiments (Supplementary Fig. 26). On the other hand, the in situ Raman results revealed the presence of a large amount of oxygen vacancies on the catalyst, which is consistent with the presence of Ce$_2$O$_3$. For the Pd/CeO$_2$ nanoparticle catalyst and Pd$_1$/CeO$_2$-AT SAC, we did not observe any enhanced oxygen activation via in situ Raman spectroscopy (Supplementary Figs. 27 and 28).

In situ HCHO-DRIFTS was carried out at 30 °C over the Pd/CeO$_2$-AT-S catalyst (Fig. 3f) to investigate the surface species and active sites in the reaction. When HCHO/N$_2$ purges the catalyst for 30 min (curve 2), we observed the gradual consumption of the hydroxyl group on the catalyst surface, accompanied with the presence of formate (HCOO) and dioxymethylene groups (DOM, H$_2$CO$_2$). The assignments of the peaks are shown in Supplementary Table 5. When switching to N$_2$, we observed an increase in the intensity of the HCOO species (Fig. 3f, curve 3), although the overall intensity of the DOM decreased and the relative intensity of the two C-O bonds changed (Supplementary Fig. 29), indicating the transition of DOM to formate in flowing N$_2$. When switching to O$_2$, we found the recurrence of the intensity of the hydroxyl group (Fig. 3f, curve 4), indicating the production of hydroxyl groups on the catalyst with the presence of O$_2$. In the presence of HCHO/O$_2$/N$_2$ (Supplementary Fig. 30), the intensities of the bands of HCOO and DOM species are lower as compared with those in the flowing HCHO/N$_2$. Likewise, the HCHO oxidation over the Pd$_1$/CeO$_2$-AT SAC follows the reaction pathways of HCHO to CO (Supplementary Fig. 31), indicating a different reaction mechanism as compared to the Ce$_2$O$_3$-Pd domain catalyst (Fig.3c).

## DFT calculations of catalyst formation and HCHO oxidation processes

Based on the above characterization results, we conclude that high-temperature steam treatment of the atom-trapped Pd$_1$/CeO$_2$ SAC generates a Ce$_2$O$_3$-Pd domain interface, which efficiently catalyzes formaldehyde oxidation at low temperatures. The generation of Ce$_2$O$_3$ is pivotal because Ce$^{3+}$ is associated with the oxygen vacancies, which can readily activate the oxygen molecules. The amount of the oxygen vacancies is proportional to the treatment time showing the first an upshift in the Raman spectral features (from 463 cm$^{-1}$) as the treatment time was increased up to 9 h and then a downshift by further steam treating the Pd$_1$/CeO$_2$-AT catalyst to 18 h. We used density functional theory (DFT) to investigate the formation process of the Ce$_2$O$_3$-Pd domain interface derived from the atom-trapped Pd$_1$/CeO$_2$ SAC. Since the Pd metal atom is more stable on a CeO$_2$ step at elevated temperatures[31], the model of the atom-trapped Pd$_1$/CeO$_2$-AT catalyst was therefore built as Pd$_1$ located on the step site of CeO$_2$ (111) (Pd$_1$/step, ii in Fig. 4a). For comparison, a model with Pd$_1$ located on the terrace site of CeO$_2$ (111) (Pd$_1$/terrace, i in Fig. 4a) was also adopted for Pd$_1$/CeO$_2$ SAC prepared via impregnation. For Pd$_1$/terrace, the binding energy of Pd$_1$ is −2.15 eV (Supplementary Fig. 32a), which is smaller than its cohesion energy (3.69 eV), and its migration on the terrace requires overcoming a small energy barrier of 0.14 eV, indicating that aggregation of Pd$_1$ on the terrace may occur. Likewise, the binding energy of Pd$_1$ on Pd$_1$/step is −3.77 eV (Supplementary Fig. 32b), which is slightly greater than the cohesion energy (3.69 eV). The energy barrier of Pd$_1$ migration on the step is as large as 1.45 eV, indicating the high stability of the atom-trapped Pd$_1$/CeO$_2$ SAC, which agrees with the results observed in the CeO$_2$-trapped Pt system[7,32].

Since the steam treatment of Pd$_1$/CeO$_2$-AT generates the highly active Pd/CeO$_2$-AT-S catalyst, we next investigate the mobility of Pd atoms on the Pd$_1$/CeO$_2$ SAC in the presence of water vapor at elevated temperatures via DFT. Firstly, the calculations show that the dissociation of H$_2$O requires low energy barriers of 0.14 and 0.04 eV on the terrace and step of the bare CeO$_2$ (Supplementary Fig. 33),

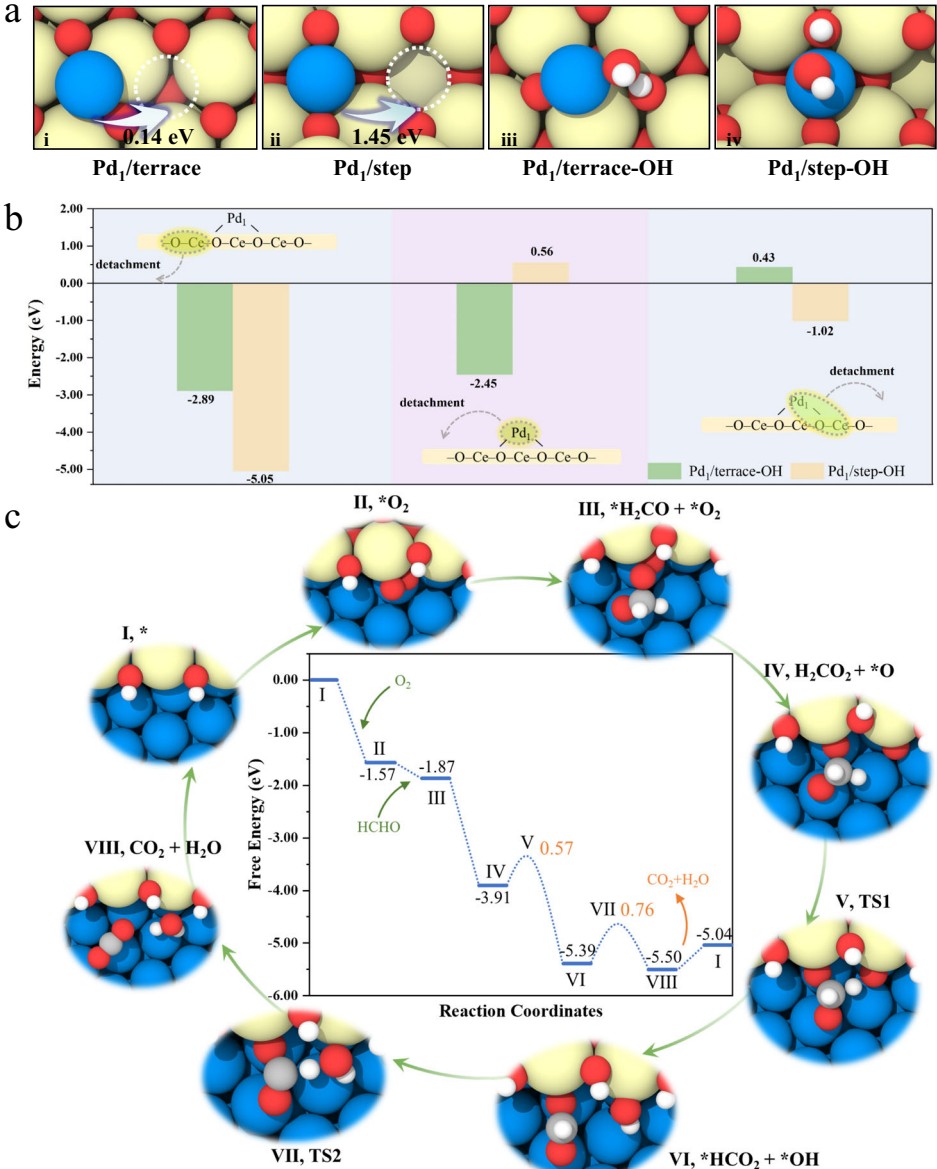

**Fig. 4 | DFT simulation demonstrating the evolution process of the Pd/CeO₂-AT-S and reaction mechanism of formaldehyde oxidation. a** Optimized structures of (i) Pd₁/terrace, (ii) Pd₁/step, (iii) final state of H₂O dissociation over Pd₁/terrace (Pd₁/terrace-OH) and (iv) final state of H₂O dissociation over Pd₁/step-OH), with migration barrier to position in white dash circle. **b** Detachment energies of Ce-O, Pd and Pd₁-O-Ce from Pd₁/terrace-OH and Pd₁/step-OH, as reference to the same detachment behavior on the bare terrace (all cases were attached in Supplementary Figs. 40–43). **c** Calculated energy profiles of HCHO oxidation (30 °C) at the Ce₂O₃-Pd domain interface, with the structures of the key intermediates and transition states (TSs). Yellow, red, gray, white, and blue circles denote Ce, O, C, H and Pd atoms, respectively.

respectively, suggesting that the surfaces of both catalysts are covered by OH and H. On Pd₁/terrace, the dissociation of H₂O has a barrier of 0.09 eV with the produced OH and H bound to Pd₁ and lattice O (Pd₁/terrace-OH, iii in Fig. 4a), respectively (Supplementary Fig. 34a). On the Pd₁/step, the dissociation of H₂O requires a barrier of 0.29 eV (Supplementary Fig. 34b). The resulting OH species adsorbed on the Pd₁ site (Pd₁/step-OH, iv in Fig. 4a) lead to a low energy barrier (0.51 eV) for Pd₁ migration (Supplementary Fig. 35), indicating that the H₂O can promote the Pd aggregation on Pd₁/step. Without the addition of the H₂O, the high-temperature treatment of Pd₁/CeO₂-AT in an inert gas (such as in Ar at 750 °C, the catalyst was denoted as Pd/CeO₂-AT-Ar) did not generate the ensemble of Pd nanoparticles and Ce³⁺ species (Supplementary Figs. 36 and 37), leading to the poor formaldehyde oxidation performance (Supplementary Fig. 38). Therefore, steam treatment is favorable for the migration of Pd and Ce atoms over the

Pd₁/CeO₂-AT sample and indeed also lead to the exchange of oxygen species between the Pd₁/CeO₂-AT and the H₂O molecule (Supplementary Fig. 39).

We further calculated the detachment energies of various catalyst structural units (e.g., Ce-O, Pd₁, and Pd₁-O-Ce) in the presence and absence of water dissociation (Fig. 4b). The detachment energy is defined in Supplementary Materials and calculated via the formula of $E_{detach} = E_{unit} + E_{defective\text{-}surface} - E_{surface}$, where $E_{unit}$, $E_{defective\text{-}surface}$ and $E_{surface}$ represent the energy of the unit (e.g., Ce-O, Pd₁ and Pd₁-O-Ce), the surface from which the unit is removed (e.g., steps and terraces) and the pristine surface, respectively. A more positive $E_{detach}$ value means that the unit is more tightly bound to the surface, i.e., more difficult to be detached from the surface. The detachment energy of the unit on the bare terrace is set to 0 eV for reference and all values and structures are shown in Supplementary Tables 7-9 and

Supplementary Figs. 40–43. Figure 4b shows that the relative detachment energy (−5.05 eV) of Ce-O from $Pd_1$/step-OH is much lower than that (−2.89 eV) on $Pd_1$/terrace-OH, suggesting that the Ce-O unit is thermodynamically less stable on $Pd_1$/step-OH. For the detachment of $Pd_1$-OH, the calculations showed that the relative detachment energy of $Pd_1$-OH on $Pd_1$/step-OH is 3.01 eV (Supplementary Table 8) higher than that on $Pd_1$/terrace-OH, indicating that the $Pd_1$-OH entity is more stable on the step than on the terrace. The improved stability of the $Pd_1$-OH on the step than on the terrace possibly indicates a higher coordination number over the $Pd_1$/$CeO_2$-AT SAC, as revealed by the EXAFS fitting (Supplementary Table 2). Finally, we investigate the detachment of $HO$-$Pd_1$-$Ce$-OH from the catalyst surfaces. The results show that the relative detachment energy of $HO$-$Pd_1$-$Ce$-OH on $Pd_1$/step-OH is 1.45 eV lower than that on $Pd_1$/terrace-OH (Fig. 4b), indicating that the direct detachment of $Pd_1$-Ce-O from $Pd_1$/step-OH is easier than that from $Pd_1$/terrace-OH. Overall, these results suggest that migration of Ce-O-based structures in the presence of $H_2O$ is more likely to occur at the step site. The scheme illustration for the formation of Pd nanoparticles and $Ce_2O_3$-Pd nanoparticle domain interface under the treatments of both Ar and steam at 750 °C is displayed in Fig. 3g.

The reaction routes of the formaldehyde oxidation at the $Ce_2O_3$-Pd interface were also investigated by DFT. The interface consists of a clipped O-Ce-O trilayer covering the surface of Pd(111), where the edge O is passivated by H species due to steam treatment and the Ce is reduced (Supplementary Table 10) relative to that in $CeO_2$ (see Supplementary Fig. 44 for the construction details of the interface structure). Figure 4c shows the calculated free energy profile of the reaction pathway at the $Ce_2O_3$-Pd interface. $O_2$ is first adsorbed at the interface (II, *$O_2$) of the bare catalyst (*), where one O is located at the Ce-Ce bridged site and bound to Pd. The other O is coordinated to Ce only (denoted as less-coordinated O), yielding an adsorption free energy of −1.57 eV. Followed by the adsorption of HCHO, O and C bonds to the Pd surface and the less-coordinated O in *$O_2$ (III, *$H_2CO_3$), respectively, generating an adsorption free energy of −0.30 eV. Subsequently, the O-O bond in *$O_2$ is spontaneously broken to produce an adsorbed DOM and an *O at the interface (IV, *$H_2CO_2$ + *O), showing no free energy barrier and releasing a reaction energy of 2.04 eV. Then, one H of the DOM attacks the *O, producing a *$HCO_2$ co-adsorbed with an *OH (VI). This step requires a free energy barrier of 0.57 eV. After that, the H in *$HCO_2$ reacts with *OH to form *$CO_2$ and *$H_2O$ (VIII), which overcame a free energy barrier of 0.76 eV. Finally, the $CO_2$ and $H_2O$ species desorb from the $Ce_2O_3$-Pd interface with a free energy of 0.46 eV to finish the reaction cycle. In the entire reaction process, the free energy curve keeps going downhill with a rate-determining step energy barrier of 0.76 eV, showing better performance than that of $Pd_1$/step (Supplementary Figs. 45 and 46). Apart from DFT calculations, the reaction order of $O_2$ from kinetic results for Pd/$CeO_2$-AT and Pd/$CeO_2$-AT-S is also an important evidence. The reaction order of $O_2$ directly reflect the distinct $O_2$ activation capability of the two catalysts. Pd/$CeO_2$-AT cannot activate the gaseous oxygen at low temperature, instead, it follows Mvk mechanism, i.e., the lattice oxygen is the active species. HCHO is oxidized by the lattice oxygen, the consumption of which is recovered by the gaseous oxygen. This oxygen activation process is slow and occur at high temperature (å 100°C). For the Pd/$CeO_2$-AT-S, due to the presence of the interface of $Ce_2O_3$-Pd, the reaction proceeds through the L-H mechanism, where gaseous oxygen is adsorbed and activated at room temperature, which is much more active than the lattice oxygen of $CeO_2$.

In summary, we report the formation of an active site in heterogeneous catalysis, the oxide-metal domain interface, generating by starting from a single-atom catalyst via the coordinated migration of metal and oxide atoms. The as-prepared $Pd_1$/$CeO_2$ single-atom catalyst by atom trapping is not active at room temperature, but after steam treatment, it catalyzes HCHO oxidation with ~100% conversion. We

show that the coordinated mobility of both Pd and Ce atoms helps to generate the $Ce_2O_3$-Pd domain interfaces, which are responsible for the enhanced activity. The oxide-metal domain interfaces which constitute the active sites were identified by several characterization techniques and corroborated by DFT simulation. Similar enhanced activity after steam treatment was also observed in toluene oxidation (Supplementary Fig. 47) or on other single-atom catalysts prepared by atom trapping, such as Ir/$CeO_2$ and Pd/$MnO_2$ (Supplementary Figs. 48 and 49). The formation of such oxide-metal domain interfaces therefore appears to be a general feature when using single-atom catalyst as a catalyst precursor. This work has set out a design principle for producing efficient catalysts by generating an oxide-metal domain interface for environmental catalysis.

## Methods
### Chemicals and materials
Cerium (III) nitrate hexahydrate (Ce($NO_3$)$_3$ · 6$H_2O$, 99.99%) was purchased from Alfa Aesar Company. Palladium nitrate dihydrate (Pd($NO_3$)$_2$ · 2$H_2O$, Analytical reagent, Pd ≥ 39.5%) and $MnO_2$ were purchased from Sinopharm Chemical Reagent Limited corporation. Hydrogen hexachloroiridate (IV) hexahydrate ($H_2IrC_{16}$.6$H_2O$, Ir≥ 35.02%) was purchased from Kunming Guiyan New Materials Technology Company Limited. Water$^{-18}$O (97 atom% $^{18}$O) was obtained from Meryer (Shanghai) Chemical Technology Limited corporation. Paraformaldehyde (Extra pure, 96%) was obtained from J&K Scientific Limited corporation. Aluminum oxide ($\gamma$-$Al_2O_3$, 99.99%) was purchased from Aladdin Company. All the reagents were used without further pretreatment.

### Preparation of supported catalysts
**Synthesis of $CeO_2$ support.** 15 g Ce ($NO_3$)$_3$·6$H_2O$ was calcined in static air at 350 °C for 2 h to obtain $CeO_2$ powder, which was used as the support to prepare the Pd/$CeO_2$ catalysts.

**Synthesis of Pd single-atom catalysts.** 1.0 wt% $Pd_1$/$CeO_2$ catalysts were prepared by impregnation (I) and atom trapping (AT). The palladium nitrate solution was prepared by dissolving 1 g of Pd($NO_3$)$_2$ · 2$H_2O$ powder in 3 mL of $HNO_3$. Then, 0.216 mL palladium nitrate dihydrate solution and 0.05 mL of deionized water were mixed, following by dropping into $CeO_2$ (1 g) dropwise while grinding in a mortar and pestle. Then the obtained mixture was dried at 80 °C for 12 h in the static air, and calcined in a muffle furnace at 500 °C and 800 °C for 12 h at a temperature ramping rate of 10 °C·min$^{-1}$. The obtained materials were denoted as $Pd_1$/$CeO_2$-I and $Pd_1$/$CeO_2$-AT, respectively.

**Steam pretreatment.** The $Pd_1$/$CeO_2$-I and $Pd_1$/$CeO_2$-AT catalysts were further treated with steam (S), denoted as Pd/$CeO_2$-I-S and Pd/$CeO_2$-AT-S, respectively. Steam treatment was conducted using a fixed bed flow reactor. 0.2 g of catalyst (40–60 mesh) was packed between two quartz wool plugs inside a quartz tube (inner diameter = 7.0 mm), and the upper part was filled with SiC to fully preheat the gas. And the catalyst was treated with 10 v/v% $H_2O$/Ar at 750 °C for 9 h at a temperature increase rate of 10 °C·min$^{-1}$, typically with a space velocity of 10,000 mL·g$^{-1}$·h$^{-1}$. The sample was then cooled down to 300 °C and the water vapor was discontinued. Afterward, the sample was purged in the same carrier gas for 1 h before cooling down to room temperature. By adjusting the steam treatment time of the catalyst, the formation process of the catalyst structure was observed. The obtained samples were denoted as Pd/$CeO_2$-AT-S x h, where x stands for the time of steam treatment (x = 0, 3, 4, 6, 9, 12, 15, 18). 10 vol% of water was injected into the flowing gas stream by a calibrated syringe pump (LSP02-1B, Longer Pump) and was vaporized in the heated gas line (200°C) before entering the reactor. The pumping rate of water was 2.45 μL·min$^{-1}$ and the total flow rate was 33.3 mL·min$^{-1}$, corresponding to the space velocity of 10,000 mL·g$^{-1}$·h$^{-1}$. Similarly, the bare $CeO_2$

powder with the same steam treatment by 10 v/v% $H_2O$/Ar at 750 °C for 9 h, was denoted as $CeO_2$-S. As a comparison, the $Pd_1/CeO_2$-AT catalyst was thermally treated with Ar (33.3 mL·min⁻¹) at 750 °C for 9 h, which was denoted as $Pd/CeO_2$-AT-Ar. 1.0 wt% $Pd/Al_2O_3$ catalyst was prepared according to the literature[33]. $Al_2O_3$ was used and has a BET surface area of 146 m²·g⁻¹. Samples were prepared by incipient wetness impregnation (IWI) with a solution of palladium nitrate dihydrate as a precursor. After impregnation, the sample was dried at 80 °C and calcined at 750 °C for 9 h with Ar. The temperature was raised at a rate of 10 °C·min⁻¹.

**Synthesis of $Pd/MnO_2$ and $Ir/CeO_2$ single-atom catalysts.** 1.0 wt% $Pd/MnO_2$ catalyst was prepared by atom trapping (AT). Briefly, 1 g of $Pd(NO_3)_2 \cdot 2H_2O$ powder was dissolved in 3 mL of $HNO_3$ and the obtained solution was mixed with deionized water to make 10 mL palladium nitrate solution. 0.216 mL palladium nitrate dihydrate solution was mixed with 0.05 mL of deionized water, which was dropped into $MnO_2$ (1 g) dropwise while grinding in a mortar and pestle. Then the obtained mixture was dried at 80 °C for 12 h in the static air, and calcined in a muffle furnace at 800 °C for 12 h at a temperature ramping rate of 10 °C·min⁻¹. The obtained catalyst was denoted as $Pd/MnO_2$. The $Pd/MnO_2$ catalyst was further treated with steam (S) at 750 °C, which was denoted as $Pd/MnO_2$-S.

1.0 wt% $Ir/CeO_2$ catalyst was also prepared by atom trapping (AT). Briefly, an appropriate amount of hydrogen hexachloroiridate (IV) hexahydrate solution was added drop-wise to the $CeO_2$ while being kept grinding in a mortar and pestle. Then, the obtained mixture was dried at 80 °C for 12 h in the static air, and calcined in a muffle furnace at 800 °C for 12 h at a temperature ramping rate of 10 °C·min⁻¹. The obtained material was denoted as $Ir/CeO_2$. The $Ir/CeO_2$ catalyst treated in steam (S) was denoted as $Ir/CeO_2$-S.

## HCHO oxidation reactivity

The activity measurement for the HCHO oxidation over the catalysts was performed in a fixed bed quartz flow reactor (inner diameter = 7.0 mm) under atmospheric pressure. Typically, 30 mg catalyst (40–60 mesh) was mixed with 170 mg quartz sand and loaded into a quartz tube reactor. The feed gas composition for the activity tests was 400 ppm HCHO/20% $O_2$/$N_2$ (total flow 50 mL·min⁻¹) at a weight hourly space velocity of 100,000 mL·g⁻¹·h⁻¹. Gaseous HCHO was generated by passing $N_2$ through a paraformaldehyde container in the thermostatic water bath, where HCHO concentration can be changed by manipulating temperature and flow rate. To investigate the moisture effect, 10% relative humidity was added by the pump. The oxidation products in the effluent gas were analyzed by an on-line gas chromatograph (GC2060, Shanghai Ruimin GC Instruments Inc.) equipped with a hydrogen flame ionization detector (FID) and Ni catalyst converter that was used for converting carbon oxides quantitatively into methane before the detector. The concentration of HCHO was calculated by the external standardization through the calibrated curve of $CO_2$.

Since no other carbonaceous compounds except $CO_2$ were detected in the effluents for all the catalysts, the HCHO conversion was expressed as Eq. (1) below, where $[CO_2]_{out}$ was the concentration of $CO_2$ produced at a certain temperature and $[CO_2]_{total}$ represented the concentration of $CO_2$ in the outlet gas when HCHO was oxidized into $CO_2$.

$$X_{HCHO} = \frac{[CO_2]out}{[CO_2]total} \times 100\% \qquad (1)$$

The reaction rate ($r_{HCHO}$) of HCHO (in a unit of µmol·g$_{Pd}$⁻¹·s⁻¹) was calculated by Eq. (2):

$$r_{HCHO} = \frac{X_{HCHO} \times f_{HCHO}}{m_{cat} \times n} \qquad (2)$$

Where $f_{HCHO}$ refers to the flow rate of HCHO (in a unit of µmol·s⁻¹), and $m_{cat}$ refers to the mass of catalyst (in a unit of g) in the fixed bed and $n$ is the weight percentage of the Pd catalysts.

## Kinetic reaction measurement

Kinetic data were determined by separate experiments, and the HCHO conversion was kept below 20% to ensure the reaction under the intrinsic kinetic regime. The effect of internal and external diffusion was also all eliminated (Supplementary Fig. 16).

For the activation energies ($E_a$) measurement, the feed gas composition of 520 ppm HCHO/20% $O_2$/$N_2$ (total flow 100 mL·min⁻¹) was applied. The activation energies were calculated by Eq. (3):

$$\ln r_{HCHO} = -\frac{E_a}{RT} + \ln A \qquad (3)$$

Where $E_a$ refers to the apparent activation energy (in a unit of J·mol⁻¹), R refers to the universal gas constant (in a unit of J·mol⁻¹·K⁻¹), and $T$ refers to the reactor temperature (in a unit of K).

Reaction rate as a function of the HCHO or $O_2$ partial pressure was measured at 30 °C or 120 °C. To determine the influence of the oxygen concentration on the oxidation reaction rate, the experiment was conducted using different atmospheres with oxygen concentrations of 5%, 10%, 15%, 20% 25%, and 30% (by volume), respectively. The feed gas composition of 400 ppm HCHO/x% $O_2$/$N_2$ (total flow 100 mL·min⁻¹) was applied. To determine the influence of the formaldehyde concentration on the oxidation reaction rate, the experiment was conducted using different atmospheres with formaldehyde concentrations of 178 ppm, 300 ppm, 400 ppm, 520 ppm, 820 ppm, 1130 ppm, and 1357 ppm (by volume), respectively. The feed gas composition of y ppm HCHO/20% $O_2$/$N_2$ (total flow 100 mL·min⁻¹) was applied.

## Catalyst characterization

X-ray diffraction (XRD) patterns of catalysts were recorded on Rigaku Ultima IV X-ray diffractometer with Cu Kα radiation (40 kV and 30 mA) at a scanning speed of 2θ = 10.0 °/min from 20° to 80° and at a scanning speed of 2θ = 0.6 °/min from 38° to 42°. The radiation used was Cu Kα with a wavelength of 0.154 nm. The XRD patterns of all samples were analyzed by PDXL II software to verify the existence of a crystalline and identify the crystal structure and grain size of the phase. The grain sizes were calculated by the Scherrer equation, D = (Kλ)/(FWHM cos θ), where D is the crystal size, λ is the wavelength of the X-ray radiation and K usually is taken as 0.89, FWHM is the full width at half maximum in radian of the diffraction peaks. X-ray photoelectron spectra (XPS) were recorded at room temperature by a Thermo Scientific K-Alpha instrument (Physical Electronics) using Al K$_\alpha$ radiation (1486.6 eV), which was operated at 12 kV and 6 mA. The pass energy was 50 eV, and the base pressure of the analysis chamber was 5.0 × 10⁻⁷ mbar. The binding energy was calibrated by using the C$_{1s}$ photoelectron peak at 284.8 eV. Analysis of the XPS spectra was performed using Thermo Avantage 5.52 Surface Chemical Analysis software. The surface area and pore volume of the samples were measured using a Micromeritics ASAP 2020 Plus instrument. The surface area analyzer according to the multi-point Brunauer Emmett Teller (BET) method with $N_2$ adsorption at −196 °C. The Pd loadings of samples were determined by Inductive Coupled Plasma-Optical Emission Spectroscopy (ICP-OES) on an Agilent 720ES system.

## X-ray Absorption Fine Structure (XAFS)

X-ray Absorption Fine Structure (XAFS) spectra at the edge were measured in transmission mode by using synchrotron radiation with a Si (111) double crystal monochromater at the EXAFS station (Beamline BL11B1) of Shanghai Synchrotron Radiation Facility (SSRF). All data were collected in fluorescence mode. To suppress the unwanted

harmonics, the angle between the monochromater crystal faces was adjusted to mistune the incident beam by 30%. The incident and output beam intensities were monitored and recorded using a nitrogen gas and a 50% argon-doped nitrogen flowing ionization chamber. The energy resolution was about 1.5 eV for XANES and about 3.0 eV for the EXAFS. Data reduction, data analysis, and EXAFS fitting were performed with the Athena and Artemis software packages[34]. The Pd K-edge ($E_0 = 24352$ eV) X-ray absorption near edge structure (XANES) and extended X-ray absorption fine structure (EXAFS) experiments were performed at 3.5 GeV under the "top-up" mode with a constant current of 240 mA. The energy calibration of the catalysts was conducted through a standard Pd foil, which as a reference was simultaneously measured. For EXAFS modeling, EXAFS of the Pd foil is fitted by fixing the coordination number (CN) of 12 for the Pd-Pd bond and the obtained amplitude reduction factor $S_0^2$ value (0.70) was set in the EXAFS analysis to determine the CN in the Pd-O scattering path in Pd/CeO$_2$ catalysts. $\Delta E_0$ was shared for all the shells, while $\sigma^2$ and $\Delta R$ values were shared within each shell for the same type of back-scattering atoms. The Ce L$_3$-edge ($E_0 = 5723$ eV) XANES and EXAFS experiments were performed at 2.2 GeV under the "top-up" mode with a constant current of 80 mA. Measurement of Ce$^{4+}$/Ce$^{3+}$ using synchrotron radiation typically exploits the Ce L$_3$-edge XANES spectra, which involves a transition of a 2p electron to an unoccupied 5d state. Depending on the oxidation state the L$_3$-edge shifts from ~5723 eV for Ce$^{3+}$ to ~5740 for Ce$^{4+}$, resulting from differences in the coulomb interaction between the 2p core hole and the valence band. The asymmetries in the L$_3$ edge and its shoulders were identified by fitting the L$_3$ curve with Gaussian and arc-tan functions. Five weak shoulders can be seen as *A, B, C, D,* and *E* (Supplementary Table 4 and Supplementary Fig. 13). At the pre-edge region the peak denoted as *A* is associated with a forbidden dipole transition from the bottom of the conduction band. The single peak (*B*) is indicative of the trivalent (Ce$^{3+}$) state, and the other peaks *C, D,* and *E* indicate the presence of the Ce$^{4+}$ oxidation state[35,36]. The estimated value of Ce$^{3+}$ concentration was taken to be the relative integrated area of peak "*B*" concerning the total area under the Ce L$_3$ curve and it can be calculated roughly using the following formula (4):

$$Ce^{3+}\% = \frac{Peak\ area\ of\ B}{Total\ area\ (B,C,D\ and\ E\ peaks)}\% \qquad (4)$$

The area under the "*A*" feature is small and ignored.

### High sensitivity low energy ion scattering (HS-LEIS)
High Sensitivity Low Energy Ion Scattering (HS-LEIS) is a surface-sensitive technique, which probes only the topmost layer of the sample[37,38]. HS-LEIS spectra were obtained using an Ion-TOF Qtac100 instrument using a scattering angle of 90°. To minimize the damage to the surface, Ne was selected as the ion source with a kinetic energy of 5 keV, an ion flux of 6000 pA·m$^{-2}$, a spot size of $2 \times 2$ mm$^2$, and an analysis time of 240 s. In this work, HS-LEIS was performed on different catalysts, and multiple sputtering was performed on the Pd/CeO$_2$-AT-S catalyst to obtain HS-LEIS spectra at different depths.

### Electron microscopy
High-angle-annular-dark-field scanning transmission electron microscopy (HAADF-STEM) images and electron energy loss spectrum (EELS) spectrums were obtained using an aberration-corrected transmission electron microscope (TEM, Thermo Fisher FEI Themis Z G3 60-300) equipped with a Schottky extreme field emission gun and a high-resolution Gatan imaging filter (GIF 994, Gatan Inc.). The acceleration voltage of the TEM was 300 kV. The energy resolution of the electron energy loss spectra is approximately 1 eV. For high loss spectrums, 0.1 eV/channel dispersion was used. STEM-EDX elemental maps were obtained using the FEI Super-X EDX detector. EELS analysis resolved

the chemical environment of Pd on ceria. The sharp peaks at 875–930 eV are generally attributed to the Ce M$_{4,5}$ edge. The chemical state of cerium could be judged from the characteristic peaks of Ce M$_{4,5}$ edge. The most obvious difference is that the weaker features of Ce$^{3+}$ occur on the higher energy sides of the main peaks, while Ce$^{4+}$ on the contrary[23].

### CO-temperature programmed reduction (CO-TPR)
The catalyst was further characterized by CO-TPR for oxygen and surface hydroxyl species for the steam-treated catalysts. H$_2$ and CO$_2$ may be produced simultaneously from the water-gas shift due to the interaction of CO with the surface hydroxyl groups in the catalysts[39–41], while CO$_2$ only is emitted through the reduction of surface lattice oxygen from Pd-O-Ce and CeO$_2$.

CO (ads) + 2OH (support) → CO$_2$ (g) + H$_2$ (g) + O$^{2-}$ (support)

CO (ads) + PdO → Pd + CO$_2$ (g)

Before the CO-TPR measurement, the Pd$_1$/CeO$_2$-AT catalyst was treated by the steamed water-$^{18}$O, which was denoted as Pd/CeO$_2$-AT-Steam H$_2^{18}$O. CO-TPR was carried out on a homemade instrument, where 100 mg catalyst was typically used in each test. Firstly, the Pd/CeO$_2$-AT-Steam H$_2^{18}$O catalyst was pretreated in a flow of He (40 mL·min$^{-1}$) at 200 °C for 1 h, then cooled down to room temperature in the He atmosphere. Next, 1% CO/He (20 mL·min$^{-1}$) was introduced. After the gas flow was stabilized, the temperature was ramped from room temperature to 500 °C at a ramping rate of 10 °C·min$^{-1}$. The evolved CO$_2$ (m/z = 44, 46) and H$_2$ (m/z = 2) were analyzed by an online Pfeiffer Omni Star GSD 320 quadrupole mass spectrometer (QMS,) equipped with a Secondary Electron Multiplier (SEM) detector.

### HCHO temperature programmed surface reaction (HCHO-TPSR)
HCHO temperature programmed surface reaction (HCHO-TPSR) experiments were performed to explore the possible reaction route. Typically, 100 mg of catalyst (40–60 mesh) was loaded in a quartz reactor and pretreated at 200 °C for 60 min under N$_2$ (30 mL·min$^{-1}$). The reactor was then cooled down to room temperature. Gaseous HCHO was generated by passing He (15 mL·min$^{-1}$) through a paraformaldehyde container in a thermostatic water bath (40 °C). After HCHO flowed through the reactor for 60 min, allowing the substances involved to adsorb on the catalyst, He was introduced into the reactor to remove the unabsorbed substances, and the temperature was ramped at 10 °C·min$^{-1}$ from room temperature to 300 °C in a flow of 2% O$_2$/He (30 mL·min$^{-1}$). The CO$_2$, CO, H$_2$, and HCHO production were analyzed online by the mass spectroscopy (Hiden QGA Gas Analysis system).

### O$_2$ temperature programmed desorption (O$_2$-TPD)
The adsorption strength of O$_2$ was determined by O$_2$ temperature programmed desorption (O$_2$-TPD). O$_2$-TPD is performed on AMI-300 flagship automatic temperature programmed chemisorption instrument, typically using 100 mg of catalyst (40–60 mesh) per test. First, the catalyst was pretreated at 200 °C under He for 60 min, then cooled down to 30 °C in the He atmosphere. Next, O$_2$ (30 mL·min$^{-1}$) was introduced for adsorption for 60 min. Subsequently, after the He gas flow stabilized, the temperature was ramped at 10 °C·min$^{-1}$ from room temperature to 600 °C. The O$_2$ desorption signal was detected by TCD.

### In situ O$_2$ pulse experiment
Before the in situ O$_2$ pulse experiment, the Pd$_1$/CeO$_2$-AT catalyst was steamed with water-$^{18}$O, which was denoted as Pd/CeO$_2$-AT-Steam H$_2^{18}$O. In situ O$_2$ pulse experiment was carried out on a homemade pulse instrument, where 0.1 g catalyst was typically used in each test. The details are described below: A mixture of HCHO/He (10 mL·min$^{-1}$) was fed to the Pd/CeO$_2$-AT-Steam H$_2^{18}$O catalyst at 30 °C until the baseline reached flat and then the $^{16}$O$_2$ (99%; 0.3 mL each time) was

introduced into the flow using an injection syringe. The chemical and isotopic compositions of the reactor effluent were measured by an online Pfeiffer Omni Star GSD 320 quadrupole mass spectrometer (QMS), equipped with a Secondary Electron Multiplier (SEM) detector. The m/z signals of 18, 20, 32, 36, 44, and 46 represent $H_2^{16}O$, $H_2^{18}O$, $^{16}O_2$, $^{18}O_2$, $C^{16}O_2$, and $C^{16}O^{18}O$, respectively.

## In situ Raman spectroscopy

In situ Raman spectra were collected using a HORIBA Jobin Yvon XploRA confocal Raman system, equipped with a 632 nm laser source and Synapse Charge Coupled Device (CCD) detector. The laser power is about 3 mW, and the spectral resolution is 2 cm⁻¹. The sample was placed in a homemade Raman cell with atmosphere and temperature was controlled to allow the reactant gas to flow over the sample surface. Before the Raman measurement, the sample was pretreated at 120 °C for 30 min under 20% $O_2/N_2$ (40 mL·min⁻¹) and purged with $N_2$ for 30 min (40 mL·min⁻¹) at the same temperature. Firstly, $N_2$ (40 mL·min⁻¹) and 20% $O_2/N_2$ (40 mL·min⁻¹) gas were purged for three minutes, respectively, and cycled three times at 120 °C. Raman spectra were recorded every three minutes. The temperature was then decreased to 25 °C. Afterward, HCHO/$N_2$ (40 mL·min⁻¹) and 20% $O_2/N_2$ (40 mL·min⁻¹) were respectively purged for three minutes and cycled three times at 25 °C, and Raman spectra were recorded every three minutes. Ex situ Raman spectra were collected using an ID Spec ARC-TIC confocal Raman system equipped with a holographic notch filter, a He-Ne laser (632.8 nm), and a CCD detector. The laser power is about 3 mW, and the spectral resolution is 2 cm⁻¹. For all Pd/$CeO_2$ catalysts, Raman tests were conducted at room temperature in static air. All the obtained spectra are displayed without smooth or baseline subtraction.

## In situ Diffuse Reflectance Infrared Fourier Transformed Spectroscopy (DRIFTS)

In situ Diffuse Reflectance Infrared Fourier Transformed Spectroscopy (DRIFTS) analysis was used to investigate the adsorption behavior of CO molecule on Pd species of the Pd catalysts before and after steam treatment[42]. Briefly, DRIFTS spectra were carried out on a commercial ThermoFisher IS50 instrument equipped with a smart collector and a liquid $N_2$ cooled MCT detector. For each experiment, the spectra and backgrounds taken were averaged from 64 scans with a resolution of 4 cm⁻¹. The procedure for DRIFTS during the CO adsorption was as follows: the catalyst was pretreated at 120 °C for 30 min under 10% $O_2/N_2$ (40 mL·min⁻¹) and purged with $N_2$ for 30 min (40 mL·min⁻¹) at the same temperature. The temperature was then decreased to 30 °C and a background spectrum was taken. Afterward, IR spectra were recorded every minute. In the experiments, in order to minimize the reduction effect of CO on the Pd oxide[16], CO/$O_2$ mixture instead of pure CO was used as the adsorption gas. 1% CO and 15% $O_2$ balanced with $N_2$ were first introduced into the cell with a flow rate of 40 mL·min⁻¹ for 30 min. Subsequently, the flow of CO was discontinued while 15% $O_2/N_2$ was kept flowing for 5 min. The procedure for DRIFTS during the HCHO oxidation reaction was as follows: the catalyst was pretreated with $N_2$ (40 mL·min⁻¹) at 200 °C for 30 min. After cooling to ambient temperature, the background spectrum was collected and the procedures have been reported in our previous work[33]. Condition 1: The gases containing either 400 ppm HCHO/$N_2$ (a total flow rate of 50 mL·min⁻¹) were then introduced into the reaction cell for the adsorption experiments. Pure $N_2$ and 20% $O_2/N_2$ were used as degassing gases, respectively. Condition 2: The gases containing either 400 ppm HCHO/20% $O_2/N_2$ (a total flow rate of 50 mL·min⁻¹) were then introduced into the reaction cell for the adsorption experiments. The IR spectra were recorded every minute. All the DRIFTS spectra were analyzed by subtracting the background.

## Density functional theory (DFT) calculations

All spin-polarized DFT calculations were carried out by using Vienna ab initio Simulation Package (VASP)[43]. The projected-augmented wave (PAW) pseudopotentials were utilized to describe the core electrons[44], while valence electrons were treated by plane waves with a kinetic energy cutoff of 450 eV. The exchange-correlation potential was treated within the generalized gradient approximation according to Perdew-Burke-Ernzerhof (PBE) functional[45]. The DFT-D3 method of Grimme with the zero-damping function was adopted for vdW-dispersion energy correction[46]. The DFT + U approach introduced by ref. [47] was adopted to improve the description of strong on-site Coulomb repulsion of Ce 4$f$ electrons. $U$-$J$ was set to 4.5 eV, which has been widely reported to be effective[48–50]. The $CeO_2$ terrace surface was modeled by a $CeO_2$ (111)-p (3 × 3) supercell, consisting of three O-Ce-O trilayers (54 O atoms and 27 Ce atoms). The size of slab is 11.48 Å × 11.48 Å and a vacuum space, which is larger than 14 Å, was employed along the z-direction. A 2 × 2 × 1 Monkhorst-Pack mesh[51] was used to sample the Brillouin zone, whose convergence has been tested. The $CeO_2$ step was modeled by $CeO_2$ (111)-p (3 × 5) supercell, consisting of three O-Ce-O trilayers with the removal of a large portion of the top O-Ce-O trilayer[31]. The size of the slab is 19.13 Å × 11.48 Å. The periodic image was separated by a vacuum space larger than 14 Å along the z-direction, containing 36 Ce atoms and 72 O atoms. The k-point mesh was set to 1 × 2 × 1 after the convergence test. In the structural optimizations, the bottom O-Ce-O trilayer was fixed while others were relaxed. The convergence criteria of energy and force were set to 10⁻⁴ eV and 0.05 eV·Å⁻¹, respectively. Transition states of reactions were obtained via climbing image nudged elastic band (CI-NEB) and Dimer approach[52,53]. The convergence criteria of energy and force were set to 10⁻⁴ eV and 0.05 eV·Å⁻¹, respectively.

Since the lattices of $CeO_2$ and cubic $Ce_2O_3$ are similar[54], the $Ce_3O_3$-Pd interface was obtained by clipping the $CeO_2$ (111)-O-Ce-O trilayer supported on the surface of Pd (111), as shown in Supplementary Fig. 44A. To ensure the thermal stability of obtained interface structure, we moved supported $CeO_2$ along the x and y directions of the Pd (111) surface to obtain a series of structures, followed by the optimization of the structures and the comparison of their energies (Supplementary Fig. 44B). Finally, the most structure of the $Ce_2O_3$-Pd domain interface was obtained, where the Ce atoms at the edge were coordinated with five lattice O atoms, less than the coordination number (7) of Ce in the $CeO_2$ (111) surface. In this optimized structure, the thickness of O-Ce-O is 3.76 Å, in good agreement with the experimental observation of 0.38 nm (Supplementary Fig. 7). The Bader charge analysis showed that the Ce at the interface was reduced relative to that of $CeO_2$ (111) (Supplementary Table 10). In addition, the edge O atoms were passivated with H because of the signal of O-H observed in the experiment.

Binding energy ($E_{binding}$) of single atom was computed by the following formula (5):

$$E_{binding} = E_{surface-Pd_1} - E_{surface} - E_{Pd_1} \qquad (5)$$

in which $E_{surface-Pd_1}$, $E_{surface}$ and $E_{Pd_1}$ is the energy of surface with $Pd_1$, surface and isolated $Pd_1$, respectively. A more negative value represents stronger binding.

In the energy curves of HCHO oxidation, Gibbs free energy ($G$) was computed by the following formula (6) via VASPKIT code:

$$G = E + E_{ZPE} + \Delta U_{0 \to T} - T \times S \qquad (6)$$

in which $E$, $E_{ZPE}$, $\Delta U_{0 \to T}$, $T$ and $S$ is the electronic energy from DFT calculation, zero-point energy, internal energy difference between 0 and T K, temperature (303.15 K) adopted in our experiment and entropy computed from vibrational frequency analysis, respectively.

Adsorption free energy of reactants was computed by the following formula (7):

$$G_{ads} = G_{surface-reactant} - G_{surface} - G_{reactant} \qquad (7)$$

in which $G_{surface-reactant}$, $G_{surface}$ and $G_{reactant}$ is the free energy of surface with absorbed reactant, surface and reactant, respectively. A more negative value represents stronger adsorption.

The detachment energy is calculated via the following formula (8):

$$E_{detach} = E_{unit} + E_{defective-surface} - E_{surface} \qquad (8)$$

where $E_{unit}$, $E_{defective-surface}$ and $E_{surface}$ represent the energy of the unit (e.g., Ce-O, $Pd_1$ and $Pd_1$-O-Ce), the surface from which the unit was removed (e.g., steps and terraces) and the pristine surface, respectively. A more positive $E_{detach}$ value means that the unit is more tightly bound to the surface, i.e., more difficult to be detached from the surface. The detachment energy of the unit on the bare terrace was set to 0 eV for reference and all values and structures are shown in Supplementary Tables 6–8 and Supplementary Figs. 40–43.

## Data availability

Additional data supporting this study's findings are available from the corresponding author upon request. Source data are provided with this paper.

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

## Acknowledgements

We thank the financial support from the National High-Level Young Talents Program and National Natural Science Foundation of China (grant Nos. 22072118 (H.X.), 22121001 (H.X., Ye.W.), 22372138 (H.X.) 22303085 (Q.W.), 22373017 (S.L.) and 21973013 (S.L.)). Funds from State Key Laboratory for Physical Chemistry of Solid Surfaces of Xiamen University are also acknowledged. Part fund was supported by the Fundamental Research Funds for the Central Universities (20720220008 (H.X.)) and by Science and Technology Projects of Innovation Laboratory for Sciences and Technologies of Energy Materials of Fujian Province (IKKEM) (HRTP-[2022]–3) (H.X.). We thank the NSFC Center for Single-Atom Catalysis (22388102, H.X.). The National Natural Science Foundation of Fujian Province, China (2020J02025 (S.L.)) are also acknowledged. S.L. thanks the "Chuying Program" for the Top Young Talents of Fujian Province. Computations were performed at the Hefei Advanced Computing Center and Supercomputing Center of Fujian.

## Author contributions

L.Z. conducted the catalyst preparation; catalytic evaluation; sample characterizations and drafted the manuscript. Q.W. and S.L. performed the theoretical calculations and helped revise the manuscript. C.D., H.P., Jiw.L. and J.H. performed STEM imaging. J.Z., X.D., W.Z. and K.Z. performed the XAS measurements and analyzed the data. L.Z., D.W. and H.Z. performed Raman measurements. Y.Z. and M.C. performed LEIS measurements. H.X. performed the model visualization. Jin.L. and Sha.W. contributed to the discussion of the reactivity results. Yo.W. and A.K.D. contributed to the discussion of all the results and helped revise the manuscript. Shu.W., S.L., Ye.W. and H.X. were responsible for the supervision, funding acquisition, project administration, review, and editing.

## Competing interests

The authors declare no competing interests.
