## [Peer Review File · Nature Communications]

Generating Active Metal/Oxide Reverse Interfaces through Coordinated Migration of Single AtomsREVIEWER COMMENTS

Reviewer #1 (Remarks to the Author):

This manuscript reported the metal/oxide reverse interfaces resulted from the coordinated migration of metal and oxide atoms are the active sites in environmental catalysis. In their findings, a Pd1/CeO₂ single-atom catalyst prepared via high temperature, which is otherwise inactive, is able to completely oxidize formaldehyde after steam treatment. The enhanced reactivity is due to the formation of a Ce₂O₃-Pd nanoparticle domain interface. The authors concluded that both Ce and Pd atoms on the atom-trapped Pd1/CeO₂ catalyst migrated during steam treatment to generate this novel active interface, which is applied to other heterogeneous catalysts as well. Therefore, this work demonstrated the formation of a new active site when using metal single-atom catalyst as precursor. Conclusively, this work also provides a completely different viewpoint for the use of single-atom catalysts in heterogeneous catalysis. This manuscript is presented very well and convincingly and is of high quality. The reviewer only has the following minor questions/queries for the authors to consider to further improve the manuscript.

1. The authors reported the generation of the oxide/metal interface from the atom-trapping single atom catalyst. The reviewer wonders if this approach applies to pre-synthesized Pd nanoparticle catalyst as well. Whether Pd nanoparticles on CeO₂ can form a Ce₂O₃/Pd structure on the CeO₂ support after a similar steam treatment ?
2. Why the gas mixture of CO/O₂/N₂ was used to investigate the difference of CO adsorption on these catalysts but not CO/N₂? Why O₂/N₂ was used to remove gaseous CO but not N₂ purging?
3. In the kinetic experiments, why does the reaction rate increase first and then decrease with the increase of HCHO partial pressure?
4. The preparation process of Pd/MnO₂ and Ir/CeO₂ should be described in the preparation of catalyst.

Reviewer #2 (Remarks to the Author):

Reject:

The catalysis of noble single atom on CeO₂ is often reported. Activation with steam treatment and/or H₂-O₂ repeated treatments is also well-known. I cannot find unique new finding in this study which can satisfy the readers of Nat. Comm.

Reviewer #3 (Remarks to the Author):

In this manuscript, Zhang et al. present a novel catalyst design strategy involving the transformation of a Pd/CeO₂ single-atom precursor catalyst into an oxide-metal domain interface through steam treatment. This catalyst exhibits remarkable activity and stability in formaldehyde oxidation. The authors employed experimental characterization techniques and conducted DFT calculations to demonstrate that the oxide-metal domain interface consists of supported Pd nanoparticles covered by Ce₂O₃ overlayers with Ce³⁺ ions and oxygen defects. They further postulate and provide evidence that the interfacial region of Ce₂O₃ and metallic Pd nanoparticles serves as the active site for aldehyde oxidation. The proposed catalyst design principle is highly inspiring and holds significant relevance to the field of heterogeneous catalysis. The DFT simulations are technically sound, and the logical flow of the work is strong. However, I do have a few concerns regarding specific issues that I will outline below, indicating the need for revision.

1. High temperature steam treatment usually causes the oxidation of transition metals into higher oxidation states. What could be the main reason that the metallic nature of the Pd nanoparticle was maintained during steam treatment?
2. In line 125, a CO adsorption peak of 1970 cm⁻¹ was observed on both Pd/CeO₂-AT-S and Pd₁/CeO₂-AT catalysts. Since this peak usually corresponds to CO bridge adsorption, does it indicate that Pd-Pd dimers are formed on Pd₁/CeO₂-AT during CO treatment?
3. In Fig. 4, the transition from structure II to III, which represents the adsorption of HCHO near O₂, is characterized by an exothermic energy change of -0.3 eV. However, the C—O distance in structure III (*H₂CO + *O₂) seems to be quite short, indicating the formation of an additional C-O bond between the C in *H₂CO and the O in *O₂. An energy barrier (possibly Eley-Rideal type) might be associated with this bond formation process. Further calculation on this step might be necessary to estimate the barrier.

Reviewer #4 (Remarks to the Author):

The manuscript reports an intriguing phenomenon in which Pd single atoms aggregate into nanoparticles following by water vapor treatment, while CeO₂ migrates onto the surface of Pd nanoparticles to form Ce₂O₃ clusters, resulting in a significant enhancement of formaldehyde (FA) oxidation catalytic activity. The authors conducted a comprehensive investigation in the migration process of CeO_x to the surface of Pd NPs and its reaction intermediates for FA oxidation using techniques such as in-situ Raman, in-situ infrared spectroscopy as well as DFT calculations. The following comments/suggestions could be carefully considered before the work to be published in Nature Communications.

1. The linear adsorption peak of CO on oxide surfaces exhibits a blue shift compared to gaseous CO. In this study, only one CO adsorption peak is observed within the range above 2143 cm⁻¹. However, it is challenging to identify whether the peak at 2163 cm⁻¹ corresponds to CO adsorption on CeO₂, Ce₂O₃, unsaturated coordination sites of Ce, or PdO. In fact, oxygen-deficient CeO₂ exhibits two distinct CO adsorption peaks, attributed to linear CO adsorption on Ce⁴⁺ and unsaturated coordinated Ce⁴⁺. In addition, CO adsorption on PdO also falls within this region. The authors should provide additional

evidence to confirm that the newly observed peak at 2163 cm^{-1} corresponds to linear CO adsorption on Ce_2O_3 .

2. In Figure 2e, it is challenging to distinguish whether the d-spacing of 0.285 nm corresponds to Ce_2O_3 or PdO. Based on the lattice spacing, I am inclined to believe that these particles are associated with PdO. A lattice fringe in a single direction is not sufficient to confirm the crystal structure of a substance. Although the authors have provided additional EELS evidence, the presence of Ce could originate from dispersed Ce atoms around Pd or CeO_2 clusters. Authors should provide lattice fringes in two directions at least and their interplanar angles to confirm such explanation. Furthermore, the FFT images should be included, which can capture diffraction spots in two directions and their angles.

3. What are the distinctions between the impregnation method and the atomic trapping method? Why is it possible to obtain single-atom Pd/ CeO_2 catalysts through the atomic trapping method and position Pd atoms at lower-energy step sites?

4. The catalyst used in the study has not undergone reduction with H_2 or sodium borohydride, and Pd in the catalyst may exist in the form of single atoms or PdO. Unfortunately, it is not directly discernible whether Pd is in the single-atom form solely through FT-EXAFS. However, the authors can obtain fine structural information about Pd by fitting the EXAFS spectra. The authors could provide EXAFS and FT-EXAFS spectra for PdO, and it is possible that the FT-EXAFS spectrum of PdO exhibits a stronger peak around 0.3 nm. Additionally, the authors should conduct a more in-depth analysis of the EXAFS fitting results to demonstrate that Pd/ CeO_2 -AT is indeed a single-atom catalyst. This is essential as it forms the basis of the authors' proposed Pd and CeO_2 migration phenomenon.

5. There are distinct infrared vibrational peaks for CO on PdO and Pd1/ CeO_2 single-atom catalysts from DRIFTS experiment, which could be an important evidence to confirm the structure of Pd/ CeO_2 -AT. However, the discussion in this aspect is not sufficiently.

6. In Figure 3c, the Raman spectra of Ce-O-Ce in Pd/ CeO_2 -AT-S catalysts exhibit the peak shift as a function of treatment time under steam. This phenomenon is due to the fact that the density of lattice oxygen vacancies is increased. However, it remains unclear why prolonging the treatment duration leads to the re-upshift of the Raman peaks.

7. The Langmuir-Hinshelwood reaction mechanism is not clearly for understanding in this work. The authors should provide the detail explanations.

8. The authors noticed that no signals for any ^{18}O species are observed in Figure S21. Is it may because surface ^{18}O atoms are consumed due to the later introduction of HCHO? From the CO-TPR, it can be seen that ^{18}O has sufficient activity to react with CO at low temperatures (30 $^\circ\text{C}$ or below) and may also react with HCHO. To demonstrate the Langmuir-Hinshelwood reaction mechanism, the authors are suggested to conduct the following programmed temperature experiments: (1) Perform the experiments in Figure S21 at lower temperatures, such as temperatures before the appearance of CO-TPR signals. (2) Conduct programmed temperature surface reaction experiments where Pd/ CeO_2 -AT and Pd/ CeO_2 -AT-S catalysts are exposed to O_2 before introducing HCHO to confirm that the oxygen adsorbed on Pd/ CeO_2 -AT-S is more active. (3) Perform O_2 -TPD experiments to test the adsorption strength of O_2 on the catalyst surface.

9. The authors emphasize the higher O_2 activation capability of the Ce_2O_3 structure. Apart from DFT calculations, the reaction order of O_2 from kinetic results for Pd/ CeO_2 -AT and Pd/ CeO_2 -AT-S is also an important evidence.

10. How about the reaction stability of the steam-treated catalyst when the steam is co-fed in the reactant?

11. Ce_2O_3 is sensitive to air. Thus, it will be difficult that Ce_2O_3 still exists after air calcination at 200 °C. The authors should provide HR-TEM or STEM images of the catalyst after air treatment.

12. Determining the Ce^{3+} content in the catalyst through XANES fitting may not be appropriated. The calculation of material content using XANES should be done through non-linear least-squares fitting. The XPS measurement may one of the proper characterizations to determine the Ce^{3+} species in CeO_2 .

Response to Reviewers' Comments

Reviewer #1

This manuscript reported the metal/oxide reverse interfaces resulted from the coordinated migration of metal and oxide atoms are the active sites in environmental catalysis. In their findings, a Pd₁/CeO₂ single-atom catalyst prepared via high temperature, which is otherwise inactive, is able to completely oxidize formaldehyde after steam treatment. The enhanced reactivity is due to the formation of a Ce₂O₃-Pd nanoparticle domain interface. The authors concluded that both Ce and Pd atoms on the atom-trapped Pd₁/CeO₂ catalyst migrated during steam treatment to generate this novel active interface, which is applied to other heterogeneous catalysts as well. Therefore, this work demonstrated the formation of a new active site when using metal single-atom catalyst as precursor. Conclusively, this work also provides a completely different viewpoint for the use of single-atom catalysts in heterogeneous catalysis. This manuscript is presented very well and convincingly and is of high quality. The reviewer only has the following minor questions/queries for the authors to consider to further improve the manuscript.

1. The authors reported the generation of the oxide/metal interface from the atom-trapping single atom catalyst. The reviewer wonders if this approach applies to pre-synthesized Pd nanoparticle catalyst as well. Whether Pd nanoparticles on CeO₂ can form a Ce₂O₃/Pd structure on the CeO₂ support after a similar steam treatment?

Response: We thank the reviewer for the suggestion. According to the reviewer's suggestion, we have synthesized Pd nanoparticles with an average size of 5-6 nm using a standard Schlenk techniques (Cargnello, M. et al. Science 2013, 341, 771). After impregnating Pd nanoparticles onto ceria, the material was calcined at 300 °C for 5 h and denoted as nano-Pd/CeO₂. The nano-Pd/CeO₂ catalyst was further treated with steam (S) and denoted as nano-Pd/CeO₂-S. It was found by XRD (Figure below) that Pd nanoparticles on the nano-Pd/CeO₂ catalyst increased from 6 nm to 85 nm after steam treatment. The CO-DRIFTS of the nano-Pd/CeO₂-S sample only presented the CO adsorption on Pd nanoparticles (Figure below). The results showed that CeO₂ and Pd have no strong interaction, and their structure changed significantly after steam treatment. Therefore, the directly loaded Pd nanoparticles cannot form a strong interaction with CeO₂ and cannot form the Ce₂O₃/Pd interface structure.

Supplementary Figure for the reviewer XRD patterns for the nano-Pd/CeO₂-S and nano-Pd/CeO₂ catalysts.

Supplementary Figure for the reviewer CO-DRIFTS spectra of the nano-Pd/CeO₂-S catalyst after exposure to a flow of CO/O₂/N₂ for 30 min and degassing in O₂/N₂ for 15 min at 30°C.

2. Why the gas mixture of CO/O₂/N₂ was used to investigate the difference of CO adsorption on these catalysts but not CO/N₂? Why O₂/N₂ was used to remove gaseous CO but not N₂ purging?

Response: We thank the reviewer for the comment. Previous works indicated that Pd oxide can be reduced in the flow of CO at room temperature (Spezzati, G. *et al.* ACS Catalysis 2017, 7, 6887). To avoid the reduction of Pd oxide by CO during the DRIFTS experiment, we therefore used the gas mixture of CO/O₂ instead of CO/N₂ to investigate the CO adsorption behavior on these two catalysts. Due to the same reason as above, we used 15% O₂/N₂ to remove gaseous CO instead of N₂ purging to avoid the reduction of the Pd catalyst by the gaseous CO.

Revisions: The following sentences have been added in the Line 2-4 of Page 33: “In the experiments, in order to minimize the reduction effect of CO on the Pd oxide¹⁶, CO/O₂ mixture instead of pure CO was used as the adsorption gas.”

3. In the kinetic experiments, why does the reaction rate increase first and then decrease with the increase of HCHO partial pressure?

Response: As stated, the HCHO oxidation at the Ce₂O₃/Pd interface follows the Langmuir-Hinshelwood (L-H) reaction mechanism that HCHO and O₂ adsorbed on the active centers with competitive adsorption. At low partial pressure of HCHO, HCHO is strongly adsorbed on the site and is not affected by the change of O₂ partial pressure. While at the high partial pressure of HCHO, the active site of O₂ adsorption is replaced by HCHO, thereby reducing its reaction rate. We have described this issue in the Line 23-26 of Page 10.

4. The preparation process of Pd/MnO₂ and Ir/CeO₂ should be described in the preparation of catalyst.

Response: We appreciate the reviewer for pointing out this issue. We have added the details of the preparation process of Pd/MnO₂ and Ir/CeO₂ in the revised manuscript.

Revisions: The following paragraph has been added at Page 24-25:

“Synthesis of Pd/MnO₂ and Ir/CeO₂ single-atom catalysts 1.0 wt% Pd/MnO₂ catalyst was prepared by atom trapping (AT). Briefly, an appropriate amount of palladium nitrate dihydrate solution was added drop-wise to the MnO₂ while being kept grinding in a mortar and pestle. Then the obtained mixture was dried at 80°C for 12 h in the static air, and calcined in a muffle furnace at 800°C for 12 h at a temperature ramping rate of 10 °C·min⁻¹. The obtained catalyst was denoted as Pd/MnO₂. The Pd/MnO₂ catalyst was further treated with steam (S), which was denoted as Pd/MnO₂-S.

1.0 wt% Ir/CeO₂ catalyst was also prepared by atom trapping (AT). Briefly, an appropriate amount of hydrogen hexachloroiridate (IV) hexahydrate solution was added drop-wise to the CeO₂ while being kept grinding in a mortar and pestle. Then the obtained mixture was dried at 80°C for 12 h in the static air, and calcined in a muffle furnace at 800°C for 12 h at a temperature ramping rate of 10 °C·min⁻¹. The obtained material was denoted as Ir/CeO₂. The Ir/CeO₂ catalyst treated in steam (S) was denoted as Ir/CeO₂-S.”

In the Line 5-7 of Page 22 of the revised manuscript, the following paragraph has been added: “Palladium nitrate dihydrate (Pd(NO₃)₂·2H₂O, Analytical reagent, Pd ≥ 39.5%) and MnO₂ were purchased from Sinopharm Chemical Reagent Limited corporation. Hydrogen hexachloroiridate (IV) hexahydrate (H₂IrCl₆·6H₂O, Ir ≥ 35.02%) was purchased from Kunming Guiyan New Materials Technology Company Limited.”

Reviewer #2

Reject:

The catalysis of noble single atom on CeO₂ is often reported. Activation with steam treatment and/or H₂-O₂ repeated treatments is also well-known. I cannot find unique new finding in this study which can satisfy the readers of Nat. Comm.

Response: We agree with the reviewer that the catalysis of noble single atom on CeO₂ has been reported in some previous works. However, the present work discovered a new oxide/metal active interface after treating a single-atom catalyst in steam, which is completely different from the previous works and has never been reported. The main novelty of the

present work is listed as follows:

1. The discovery of the new oxide/metal active interface

In heterogeneous catalysis, previous works have occasionally identified metal/oxide interface as the catalytic active sites. However, the present work discovered that the REVERSE interface is the active site, i.e. oxide/metal interface. This finding is brand new and has never been reported.

2. Inactive single-atom catalyst is used as the catalyst precursor for preparing the active heterogeneous catalyst.

We found that single-atom catalyst is not always active as nanoparticle. The present work provides a new concept that the inactive single-atom catalyst can be used as catalyst precursor to prepare the active heterogeneous catalyst.

Reviewer #3

In this manuscript, Zhang et al. present a novel catalyst design strategy involving the transformation of a Pd/CeO₂ single-atom precursor catalyst into an oxide-metal domain interface through steam treatment. This catalyst exhibits remarkable activity and stability in formaldehyde oxidation. The authors employed experimental characterization techniques and conducted DFT calculations to demonstrate that the oxide-metal domain interface consists of supported Pd nanoparticles covered by Ce₂O₃ overlayers with Ce³⁺ ions and oxygen defects. They further postulate and provide evidence that the interfacial region of Ce₂O₃ and metallic Pd nanoparticles serves as the active site for aldehyde oxidation. The proposed catalyst design principle is highly inspiring and holds significant relevance to the field of heterogeneous catalysis. The DFT simulations are technically sound, and the logical flow of the work is strong. However, I do have a few concerns regarding specific issues that I will outline below, indicating the need for revision.

1. High temperature steam treatment usually causes the oxidation of transition metals into higher oxidation states. What could be the main reason that the metallic nature of the Pd nanoparticle was maintained during steam treatment?

Response: We agree with the reviewer that high-temperature steam is a weak oxidant agent. As mentioned by the reviewer, the present work indicated that the metallic nature of the Pd nanoparticle was maintained during steam treatment. This is explained by the following facts. The previous work (ChemCatChem 2021,13, 4133) has demonstrated that the decomposition temperature of PdO to Pd is ~750 °C in the 20mbar O₂ based on the function (please see the figure below) between equilibrium oxygen pressure and temperature (H. G. W. G. Bayer, Thermochimica Acta 1975, 11, 79). Without the presence of O₂, the most stable Pd phase is therefore Pd metal (please see the figure below), which was also confirmed experimentally (ChemCatChem 2021,13, 4133).

Supplementary Figure for the reviewers The diagram of PdO \leftrightarrow Pd phase transformation depicted from the function between the partial pressure of oxygen and the temperature (ChemCatChem 2021,13, 4133).

2. In line 125, a CO adsorption peak of 1970 cm^{-1} was observed on both Pd/CeO₂-AT-S and Pd₁/CeO₂-AT catalysts. Since this peak usually corresponds to CO bridge adsorption, does it indicate that Pd-Pd dimers are formed on Pd₁/CeO₂-AT during CO treatment?

Response: We thank the reviewer for pointing out this issue. Previous works indicated that Pd oxide can be reduced in the present of CO at room temperature (Spezzati, G. et al. ACS Catalysis 2017, 7, 6887). Therefore, in the CO-DRIFTS experiment, some Pd single atoms were still reduced after 30 min of CO adsorption and therefore the adsorption peak at 1970 cm^{-1} was also observed.

3. In Fig. 4, the transition from structure II to III, which represents the adsorption of HCHO near O₂, is characterized by an exothermic energy change of -0.3 eV. However, the C-O distance in structure III (*H₂CO + *O₂) seems to be quite short, indicating the formation of an additional C-O bond between the C in *H₂CO and the O in *O₂. An energy barrier (possibly Eley-Rideal type) might be associated with this bond formation process. Further calculation on this step might be necessary to estimate the barrier.

Response: We thank the reviewer for this suggestion. First, it should be noted that the direct adsorption energy for HCHO from DFT is -0.94 eV, suggesting that the interaction is not weak. The -0.30 eV in Fig. 4c of the original manuscript includes other contributions such as entropy and zero-point energy. To address the reviewer's comment, we have examined the energy barrier of HCHO adsorption via both LH and ER mechanisms. First, we simulated the coupling between the *HCHO and *O₂ following the L-H mechanism. As shown in the **Figure a** below, the HCHO adsorbed at the Pd site (3.39 Å above the Pd) encounters an energy barrier of 0.32 eV during its coupling with *O₂. For the physically adsorbed HCHO (6.33 Å above Pd), the free energy is shifted downwards during its coupling to *O₂ (E-R mechanism) and it encounters no barrier (**Figure b** below). Overall, the coupling between HCHO and *O₂ via the L-H mechanism is 0.32 eV, which is smaller than that of the other two steps (0.57 and 0.76 eV) of HCHO oxidation, while that via the E-R mechanism has no barrier.

Furthermore, we have carried out the Constrained *Ab initio* Molecular Dynamics

(Constrained AIMD) to obtain the enhanced sampling of the coupling between gaseous HCHO and *O_2 (**Figure c** below). Starting from the initial configuration that the C site in HCHO is far from 4 Å to *O_2 , the result shows that with the decrease of the C-O distance (adopted collective variable), HCHO tends to interact with Pd first and then to couple with *O_2 . The whole process encounters a barrier of about 0.40 eV, indicating the same conclusion as that obtained by CI-NEB. It should be noted that the difference between the two barriers (0.40 vs. 0.32 eV) is due to the higher energy of the TS (-CH₂ point to Pd while O point to Pd in the TS obtained by CI-NEB) obtained by constrained AIMD. Overall, the HCHO adsorption would not be the rate-determining step of the whole reaction cycle.

Supplementary Figure for the reviewer (a) Energy profile of coupling between *HCHO with *O_2 . (b) The simulated path of gaseous HCHO adsorption by CI-NEB with linear interpolation. (c) Calculated energy profile of gaseous HCHO adsorption by the constrained AIMD.

Reviewer #4

The manuscript reports an intriguing phenomenon in which Pd single atoms aggregate into nanoparticles following by water vapor treatment, while CeO_2 migrates onto the surface of Pd nanoparticles to form Ce_2O_3 clusters, resulting in a significant enhancement of formaldehyde (FA) oxidation catalytic activity. The authors conducted a comprehensive investigation in the migration process of CeO_x to the surface of Pd NPs and its reaction intermediates for FA oxidation using techniques such as in-situ Raman, in-situ infrared spectroscopy as well as DFT calculations. The following comments/suggestions could be carefully considered before the work to be published in Nature Communications.

1. The linear adsorption peak of CO on oxide surfaces exhibits a blue shift compared to

gaseous CO. In this study, only one CO adsorption peak is observed within the range above 2143 cm^{-1} . However, it is challenging to identify whether the peak at 2163 cm^{-1} corresponds to CO adsorption on CeO_2 , Ce_2O_3 , unsaturated coordination sites of Ce, or PdO. In fact, oxygen-deficient CeO_2 exhibits two distinct CO adsorption peaks, attributed to linear CO adsorption on Ce^{4+} and unsaturated coordinated Ce^{4+} . In addition, CO adsorption on PdO also falls within this region. The authors should provide additional evidence to confirm that the newly observed peak at 2163 cm^{-1} corresponds to linear CO adsorption on Ce_2O_3 .

Response: We thank the reviewer for the comments. Previous works have indicated that the CO adsorption peaks on PdO, Pd metal or Pd^{2+} were all within the range of 1800-2150 cm^{-1} (Jing et al. ACS Catalysis 2012, 2, 261. Jiang, D. et al. ACS Catalysis 2020, 10, 11356.). The CO adsorption peaks on Ce species have also been studied in some previous works (Tereshchenko, A. et al. Catalysts 2019, 9, 385.). From the literature (Tereshchenko, A. et al. Catalysts 2019, 9, 385. Konsolakis, M. Applied Catalysis B Environmental 2016. Chen, A. et al. Nature Catalysis 2019, 2, 334. C. Binet et al., Catalysis Today, 1999, 50, 207), the peak at 2172 cm^{-1} is attributed to the "top" CO adsorption on Ce^{4+} with coordination unsaturation. The peak near 2159 cm^{-1} is attributed to the adsorption of Ce^{4+} by different coordination unsaturation (Ce^{4+} cus). The CO peaks on ceria oxide observed at 2161, 2162 and 2156 cm^{-1} are attributed to the CO adsorption peaks on Ce^{3+} . To address the reviewer's comment, we have added more references in Line 21 Page 6 to emphasize this issue.

2. In Figure 2e, it is challenging to distinguish whether the d-spacing of 0.285 nm corresponds to Ce_2O_3 or PdO. Based on the lattice spacing, I am inclined to believe that these particles are associated with PdO. A lattice fringe in a single direction is not sufficient to confirm the crystal structure of a substance. Although the authors have provided additional EELS evidence, the presence of Ce could originate from dispersed Ce atoms around Pd or CeO_2 clusters. Authors should provide lattice fringes in two directions at least and their interplanar angles to confirm such explanation. Furthermore, the FFT images should be included, which can capture diffraction spots in two directions and their angles.

Response: We thank the reviewer for the professional comment on the STEM image. We agree with the reviewer that it is perfect to have the lattice fringes in two directions to distinguish the substance. Unfortunately, we have failed to find the other lattice fringe after spending much time on the microscopy, possibly due to the thin thickness or less crystallization of the Ce_2O_3 . As mentioned by the reviewer, the EELS technique is more sensitive for the detection of the Ce^{3+} species and has unambiguously confirmed the presence of Ce^{3+} . We found that the Ce^{3+} is not originated from the dispersed Ce atoms around Pd or CeO_2 clusters because no Ce^{3+} signal was detected at the places around Pd or CeO_2 after several EELS testing.

3. What are the distinctions between the impregnation method and the atomic trapping method? Why is it possible to obtain single-atom Pd/ CeO_2 catalysts through the atomic trapping method and position Pd atoms at lower-energy step sites?

Response: We thank the reviewer for the comment. The distinctions between the impregnation method and the atomic trapping method is briefly that the former was prepared at 450 °C, while the latter was prepared at 800 °C. Via atom trapping method (Jones, J. et al.

Science 2016, 353, 150.), the Pd atom is strongly bonded on the CeO₂ step site to form single-atom Pd/CeO₂ catalysts. This issue has been discussed in previous work (Jiang, D. et al. ACS Catalysis 2020, 10, 11356.), the single-atom Pd/CeO₂ catalyst was successfully prepared by the atom trapping method, which agrees well with our results that a single-atom Pd₁/CeO₂-AT catalyst was verified by CO-DRIFTS and EXAFS. Previous work has reported that Pt single atom located on the step position is more stable in the Pt₁/CeO₂ system (Pereira-Hernandez, X. I. et al. Nature communications 2019, 10, 1358. Jones, J. et al. Science 2016, 353, 150.). Using this model, we built the Pd₁/CeO₂ structure, and compared the binding energy and migration energy of Pd in the single-atom model of the step position and terrace position. We found that Pd is more stable in the step position than in the terrace (**Supplementary Table below**). Combined with the experimental results, we believe that the single-atom Pd located at the CeO₂ step is the model of the Pd₁/CeO₂-AT catalyst prepared via the atomic capture method (Kunwar, D. et al. ACS Catalysis 2019, 9, 3978.).

Supplementary Table for the reviewer (Supplementary Table 6) Different structural models for Pd₁/step and Pd₁/terrace

	Pd ₁ /CeO ₂ -AT	Pd ₁ /CeO ₂ -I
Structural model			Pd ₁ /step	Pd ₁ /terrace
Pd-O bond length/Å	2.02, 2.02	2.40, 2.40, 2.40, 2.88
Pd binding energy/eV	-3.77	-2.15
Migration model		Migration barrier /eV	1.45	0.14

Supplementary Note: As reported in the previous work, the Pt located on the step site is more stable in the Pt₁/CeO₂ system (Pereira-Hernandez, X. I. et al. Nature communications 2019, 10, 1358. Jones, J. et al. Science 2016, 353, 150.). We therefore used this structure to perform the Pd₁/CeO₂ model and compared the binding energy and migration energy of Pd

atom in the single-atom model of the step position and terrace position. We found that Pd is more stable in the step position than in the terrace (Supplementary Table 6). Combined with the experimental results, the Pd single-atom located at the CeO₂ step is regarded as the model of the Pd₁/CeO₂-AT catalyst prepared via the atom trapping method (ACS Catalysis 2019, 9, 3978).

Revisions: In the revised Supplementary Information, we have added the Supplementary Table above as the Supplementary Table 6 (Page S39) to demonstrate the different structural models for Pd₁/step and Pd₁/terrace and have added the relative description.

4. The catalyst used in the study has not undergone reduction with H₂ or sodium borohydride, and Pd in the catalyst may exist in the form of single atoms or PdO. Unfortunately, it is not directly discernible whether Pd is in the single-atom form solely through FT-EXAFS. However, the authors can obtain fine structural information about Pd by fitting the EXAFS spectra. The authors could provide EXAFS and FT-EXAFS spectra for PdO, and it is possible that the FT-EXAFS spectrum of PdO exhibits a stronger peak around 0.3 nm. Additionally, the authors should conduct a more in-depth analysis of the EXAFS fitting results to demonstrate that Pd/CeO₂-AT is indeed a single-atom catalyst. This is essential as it forms the basis of the authors' proposed Pd and CeO₂ migration phenomenon.

Response: We thank the reviewer for the comments. According to the reviewer comments, we have provided the EXAFS spectrum of PdO and performed the in-depth analysis of the EXAFS fitting results (please check the Figure below). It demonstrated that the Pd/CeO₂-AT is indeed a single-atom catalyst. We have made the revisions accordingly.

Supplementary Figure for the reviewer (Supplementary Fig. 2). Pd K-edge XANES (a) and the k₃-weighted FT-EXAFS spectra in the R-space (b) of the Pd-foil, PdO, Pd₁/CeO₂-AT and Pd₁/CeO₂-I catalysts (Fitting details are reported in Supplementary Table 2).

Revisions: We have redrawn the Figure “Pd K-edge XANES (a) and the k₃-weighted FT-EXAFS spectra in the R-space (b) of the Pd-foil, PdO, Pd₁/CeO₂-AT and Pd₁/CeO₂-I catalysts” and relative description into the revised Supplementary Information (Supplementary Fig. 2), as shown above.

5. There are distinct infrared vibrational peaks for CO on PdO and Pd₁/CeO₂ single-atom

catalysts from DRIFTS experiment, which could be an important evidence to confirm the structure of Pd/CeO₂-AT. However, the discussion in this aspect is not sufficiently.

Response: We thank the reviewer for the comment. In one of the coauthor's previous work (ACS Catalysis 2020, 10, 11356), we have carried out the detailed analysis on the DRIFTS experiment of the Pd₁/CeO₂ single-atom catalyst. This is the reason that we did not discussed it in detail in the initial submission of this manuscript. According to the reviewer's suggestion, we have appended the CO-DRIFTS results for the Pd₁/CeO₂-AT catalyst at 125°C in the revised supplementary materials as Supplementary Fig. 3, which is shown below. It is seen that for the Pd₁/CeO₂-AT catalyst, there are CO peaks at 2136 cm⁻¹ corresponding to the CO on the isolated Pd²⁺ (i.e., SAs). However, the peak of the CO bridged adsorption on Pd (< 2000 cm⁻¹) appeared only after 5 min, which was because some Pd single atoms were reduced after 20 min of CO adsorption. This phenomenon is consistent with the previous results (ACS Catalysis 2020, 10, 11356). In addition, based on the EXAFS analysis (Supplementary Fig. 2), we can further confirm that the Pd species on the Pd₁/CeO₂-AT catalyst are single atoms.

Supplementary Figure for the reviewer (Supplementary Fig. 3) CO-DRIFTS spectra of the Pd/CeO₂-AT catalyst after exposure to a flow CO/O₂/N₂ for 20 min at 125°C.

Supplementary Note: As shown in Supplementary Fig. 3, for the Pd₁/CeO₂-AT catalyst, there are initial CO adsorption peaks of 2146 and 2136 cm⁻¹ on the isolated Pd²⁺ (i.e., SAs). However, the peak value of CO-Pd₂/Pd₃ adsorption (<2000 cm⁻¹) appeared only after 5 min, which was because some single atom Pd was still reduced after 20 min of CO adsorption. This phenomenon is consistent with the previous report⁵. In addition, based on XAS analysis (Supplementary Fig. 2), we can further confirm that the Pd on the Pd₁/CeO₂-AT catalyst was a single atom.

Revisions:

We have added the Figure “CO-DRIFTS spectra of the Pd/CeO₂-AT catalyst after exposure to a flow CO/O₂/N₂ for 20 min at 125°C” and the relative description into the revised Supplementary Information (Supplementary Fig. 3).

In the Line 19 of Page 5 of the revised manuscript, we have added the following text: “XRD pattern of the Pd₁/CeO₂ SAC prepared by atom-trapping (Pd₁/CeO₂-AT) only shows

the diffraction peaks of CeO₂ support because the Pd species are atomically dispersed (Supplementary Fig. 2 and Supplementary Fig. 3).”

6. In Figure 3c, the Raman spectra of Ce-O-Ce in Pd/CeO₂-AT-S catalysts exhibit the peak shift as a function of treatment time under steam. This phenomenon is due to the fact that the density of lattice oxygen vacancies is increased. However, it remains unclear why prolonging the treatment duration leads to the re-upshift of the Raman peaks.

Response: This is a very good comment. Yes, the initial peak shift is due to the fact that the density of lattice oxygen vacancies is increased. However, prolonging the treatment duration leads to the decreased surface area of CeO₂ and the growth of Pd nanoparticles due to the low thermal stability of CeO₂ support. When the steam treatment time increases to 18 h, the Pd nanoparticles increase to ~28 nm and the average CeO₂ size increase to ~30 nm (Supplementary Fig. 18). This means that there are less Ce₂O₃ species than at 9 h treatment at this time. Therefore, when the steam treatment time exceeds 9 h, the Raman peak will re-upshift.

Revisions: In the Line 5-7 of page 12 of the revised manuscript, we have added “The upshift of the Ce-O-Ce after 9 h treatment is attributed to the decreased surface area of the catalyst and the growth of Pd nanoparticles due to low thermal stability of ceria.”

7. The Langmuir-Hinshelwood reaction mechanism is not clearly for understanding in this work. The authors should provide the detail explanations.

Response: We apologize for the missing of this information. For the Pd/CeO₂-AT-S, the reaction proceeds through the Langmuir-Hinshelwood mechanism, where gaseous oxygen is adsorbed and activated at room temperature, which reacted with the adsorbed HCHO. To address the reviewer’s comment, we have described the kinetic experiments to explain the reaction mechanism at Line 1-8 of Page 11. We have also supplemented the relative description at Line 8-11 of Page 11 of the revised manuscript.

Revisions: In the Line 4-6 of Page 11 of the revised manuscript, we have added “Therefore, the oxidation of HCHO on the Pd/CeO₂-AT-S catalyst follows the Langmuir-Hinshelwood (L-H) reaction mechanism of the competing adsorption reaction of the two adsorbed reactants²⁶.”

8. The authors noticed that no signals for any ¹⁸O species are observed in Figure S21. Is it may because surface ¹⁸O atoms are consumed due to the later introduction of HCHO? From the CO-TPR, it can be seen that ¹⁸O has sufficient activity to react with CO at low temperatures (30 °C or below) and may also react with HCHO. To demonstrate the Langmuir-Hinshelwood reaction mechanism, the authors are suggested to conduct the following programmed temperature experiments: (1) Perform the experiments in Figure S21 at lower temperatures, such as temperatures before the appearance of CO-TPR signals. (2) Conduct programmed temperature surface reaction experiments where Pd/CeO₂-AT and Pd/CeO₂-AT-S catalysts are exposed to O₂ before introducing HCHO to confirm that the oxygen adsorbed on Pd/CeO₂-AT-S is more active. (3) Perform O₂-TPD experiments to test

the adsorption strength of O₂ on the catalyst surface.

Response: We thank the reviewer for the careful reading of the manuscript. Yes, no signals for any ¹⁸O species are observed in Figure S21. This is because the reaction over the catalyst mainly follows the L-H mechanism, not the MvK mechanism in which lattice oxygen participates. According to the reviewer's suggestions, we have performed the HCHO-TPSR and O₂-TPD experiments. These experiments demonstrated that Pd/CeO₂-AT-S catalyst had stronger oxygen activation capacity and adsorption strength through O₂-TPD (Supplementary Fig. 25) and HCHO-TPSR experiments (Supplementary Fig. 26). The experiment in Supplementary Fig. 24 was performed at 30°C, which is unfortunately the lowest temperature the instrument can perform.

HCHO temperature programmed surface reaction (HCHO-TPSR) experiments were performed to explore the possible reaction route. Typically, 100 mg of catalyst (40 - 60 mesh) was loaded in a quartz reactor and pretreated at 200°C for 60 min under N₂ (30 mL·min⁻¹). The reactor was then cooled down to room temperature. Gaseous HCHO was generated by passing He (15 mL·min⁻¹) through a paraformaldehyde container in a thermostatic water bath (40°C). The HCHO flowed through the reactor for 60 min, allowing the substances involved to adsorb on the catalyst. Next, He was introduced into the reactor to remove the unadsorbed substances, and the temperature was ramped at 10 °C·min⁻¹ from room temperature to 300°C in a flow of 2% O₂/He (30 mL·min⁻¹). The CO₂, CO, H₂, and HCHO production were analyzed online by the mass spectrum (Hiden QGA Gas Analysis system). On the Pd₁/CeO₂-AT catalyst (Supplementary Fig. 26a), as the temperature increases to 42°C, most of the adsorbed formaldehyde begins to desorb, and a certain amount of CO and H₂ is also detected, probably due to the direct dehydrogenation pathway (HCHO → H₂ + CO), and it is not until 80°C that a few CO₂ production from oxidation of surface HCHO starts to occur. An amount of CO and H₂ was also detected, probably due to the direct dehydrogenation route (HCHO → H₂ + CO). By contrast, the production of CO₂ took place even at room temperature over Pd/CeO₂-AT-S catalyst, no obvious CO and H₂ were observed over the entire temperature range studied (Supplementary Fig. 26b). However, on the Pd/CeO₂-AT-S catalyst, the CO₂ signal is much lower than that on the Pd₁/CeO₂-AT catalyst. This is because most of the HCHO is oxidized at room temperature, before the introduced O₂/He reaches a stable baseline. The drastic enhancement of reactivity brought by steam treatment is in good agreement with the earlier activity tests shown in Fig. 1a.

Supplementary Figure for the reviewer (Supplementary Fig. 26). HCHO-TPSR profiles over Pd₁/CeO₂-AT (a) and Pd/CeO₂-AT-S (b) catalysts.

Supplementary Note: HCHO temperature programmed surface reaction (HCHO-TPSR) experiments were performed to evaluate HCHO oxidation as a function of temperature (30-300°C) on Pd₁/CeO₂-AT and Pd/CeO₂-AT-S, respectively. On the Pd₁/CeO₂-AT catalyst (Supplementary Fig. 26a), as the temperature increases to 42°C, most of the adsorbed formaldehyde begins to desorb, and a certain amount of CO and H₂ is also detected, probably due to the direct dehydrogenation pathway (HCHO → H₂ + CO), and it is not until 80°C that a few CO₂ production from oxidation of surface HCHO starts to occur. Both CO and H₂ were also detected, probably due to the direct dehydrogenation route (HCHO → H₂ + CO). By contrast, the production of CO₂ took place even at room temperature over Pd/CeO₂-AT-S catalyst, no obvious CO and H₂ were observed over the entire temperature range studied (Supplementary Fig. 26b). However, on the Pd/CeO₂-AT-S catalyst, the CO₂ signal is much lower than that on the Pd₁/CeO₂-AT catalyst. This is because most of the HCHO is oxidized at room temperature, before the introduced O₂/He reaches a stable baseline. The drastic enhancement of reactivity brought by steam treatment is in good agreement with the earlier activity tests shown in Fig. 1a.

O₂ temperature programmed desorption (O₂-TPD) was performed on AMI-300 flagship automatic temperature programmed chemisorption instrument to investigate the adsorption strength of O₂. In the experiment, the catalyst was pretreated at 200°C under He for 60 min, then cooled down to 30°C in the He atmosphere, followed by the O₂ (30 mL·min⁻¹) adsorption for 60 min. Subsequently, after the He gas flow stabilized, the temperature was ramped at 10 °C·min⁻¹ from room temperature to 600°C. The O₂ desorption signal was detected by TCD. For the Pd₁/CeO₂-AT, the desorption peaks of surface oxygen are located at 300°C and 450°C. Likewise, for the Pd/CeO₂-AT-S catalyst, the surface reactive oxygen species starts to gradually desorb at 30 °C. It indicated that the Pd/CeO₂-AT-S catalyst was more likely to activated O₂, which plays a key role to fulfill the catalytic oxidation of HCHO at low temperatures.

Supplementary Figure for the reviewer (Supplementary Fig. 25). O₂ temperature-programmed reduction (O₂-TPD) profiles of the Pd/CeO₂-AT-S and Pd₁/CeO₂-AT catalysts.

Supplementary Note: As for Pd₁/CeO₂-AT, the desorption peaks of surface oxygen are concentrated at

300 °C and 450 °C. Likewise, for the Pd/CeO₂-AT-S catalyst, the surface reactive oxygen species begin to be gradually desorbed at 30°C. It indicated that the Pd/CeO₂-AT-S catalyst was more likely to adsorb activated O₂, which should play a key role to fulfill the catalytic oxidation of HCHO at low temperatures.

Revisions:

In the revised Supplementary Information, we have added the above two Figures of HCHO-TPSR and O₂-TPD experiments as Supplementary Fig. 26 and Supplementary Fig. 25.

In the Line 14-26 of Page 31 of the revised manuscript, we have added “HCHO temperature programmed surface reaction (HCHO-TPSR) experiments were performed to explore the possible reaction route. Typically, 100 mg of catalyst (40 - 60 mesh) was loaded in a quartz reactor and pretreated at 200 °C for 60 min under N₂ (30 mL·min⁻¹). The reactor was then cooled down to room temperature. Gaseous HCHO was generated by passing He (15 mL·min⁻¹) through a paraformaldehyde container in a thermostatic water bath (40 °C). After HCHO flowed through the reactor for 60 min, allowing the substances involved to adsorb on the catalyst, He was introduced into the reactor to remove the unadsorbed substances, and the temperature was ramped at 10 °C·min⁻¹ from room temperature to 300°C in a flow of 2% O₂/He (30 mL·min⁻¹). The CO₂, CO, H₂, and HCHO production were analyzed online by the mass spectroscopy (Hiden QGA Gas Analysis system).”

In the Line 1-8 of Page 32 of the revised manuscript, we have added “The adsorption strength of O₂ was determined by O₂ temperature programmed desorption (O₂-TPD). O₂-TPD is performed on AMI-300 flagship automatic temperature programmed chemisorption instrument, typically using 100 mg of catalyst (40-60 mesh) per test. First, the catalyst was pretreated at 200°C under He for 60 min, then cooled down to 30°C in the He atmosphere. Next, O₂ (30 mL·min⁻¹) was introduced for adsorption for 60 min. Subsequently, after the He gas flow stabilized, the temperature was ramped at 10 °C·min⁻¹ from room temperature to 600°C. The O₂ desorption signal was detected by TCD.”

In the Line 8-10 of Page 13 of the revised manuscript, we have added “In addition, we found that Pd/CeO₂-AT-S catalyst had stronger oxygen activation capacity and adsorption strength through O₂-TPD (Supplementary Fig. 25) and HCHO-TPSR experiments (Supplementary Fig. 26).”

9. The authors emphasize the higher O₂ activation capability of the Ce₂O₃ structure. Apart from DFT calculations, the reaction order of O₂ from kinetic results for Pd/CeO₂-AT and Pd/CeO₂-AT-S is also an important evidence.

Response: We are grateful to the reviewer for this suggestion. Yes, apart from DFT calculations, the reaction order of O₂ from kinetic results for Pd/CeO₂-AT and Pd/CeO₂-AT-S is also an important evidence. The reaction order of O₂ directly reflect the distinct O₂ activation capability of the two catalysts. Pd/CeO₂-AT can't activate the gaseous oxygen at low temperature, instead, it follows Mvk mechanism, i.e., the lattice oxygen is the active species. HCHO is oxidized by the lattice oxygen, the consumption of which is recovered by the gaseous oxygen. This oxygen activation process is quite “inert”, which occurs at high temperature (> 100°C). For the Pd/CeO₂-AT-S, due to the interface of Ce₂O₃-Pd, the reaction

proceeds through the L-H mechanism, where gaseous oxygen is adsorbed and activated at room temperature, which is much more active than the lattice oxygen of CeO₂. We have added the relative description in the revised manuscript.

Revisions:

In the Line 24-26 of Page 19 of the revised manuscript, we have added “Apart from DFT calculations, the reaction order of O₂ from kinetic results for Pd/CeO₂-AT and Pd/CeO₂-AT-S is also an important evidence. The reaction order of O₂ directly reflect the distinct O₂ activation capability of the two catalysts. Pd/CeO₂-AT cannot activate the gaseous oxygen at low temperature, instead, it follows Mvk mechanism, i.e. the lattice oxygen is the active species. HCHO is oxidized by the lattice oxygen, the consumption of which is recovered by the gaseous oxygen. This oxygen activation process is slow and occur at high temperature (> 100°C). For the Pd/CeO₂-AT-S, due to the presence of the interface of Ce₂O₃-Pd, the reaction proceeds through the L-H mechanism, where gaseous oxygen is adsorbed and activated at room temperature, which is much more active than the lattice oxygen of CeO₂.”

10. How about the reaction stability of the steam-treated catalyst when the steam is co-fed in the reactant?

Response: As suggested by the reviewers, we have performed the experiments by measuring the HCHO oxidation under different humidity to evaluate the reaction stability of the steam-treated catalyst when the steam is co-fed in the reactant. As shown below, the steam-treated catalyst is stable in the 3h run under 10% humidity. We have added the relative description in the revised manuscript.

Supplementary Figure for the reviewers (Supplementary Fig. 1). Relative humidity effect on the activity of Pd/CeO₂-AT-S catalyst at 30°C. Reaction conditions: 400 ppm HCHO, 20 vol% O₂, and N₂ as balance gas, total flow rate: 50 mL·min⁻¹ and WHSV: 176,000 mL·g⁻¹·h⁻¹.

Revisions:

In the revised Supplementary Information, we have added the above figure as the Supplementary Fig. 1.

In the Line 22-26 of Page 4 of the revised manuscript, we have added “It is important to study the effect of relative humidity (RH) on the catalytic oxidation of HCHO at room temperature. Under 10% relative humidity (Supplementary Fig. 1), the reactivity of Pd/CeO₂-AT-S catalyst remained stable. It indicates that the Pd/CeO₂-AT-S catalyst has good

tolerance under humidity conditions.”

In the Line 21-22 of Page 25 of the revised manuscript, we have added: “To investigate the moisture effect, 10% relative humidity was added by the pump.”

11. Ce_2O_3 is sensitive to air. Thus, it will be difficult that Ce_2O_3 still exists after air calcination at 200 °C. The authors should provide HR-TEM or STEM images of the catalyst after air treatment.

Response: We thank the reviewer for the comment. Yes, Ce_2O_3 still exists after air calcination at 200 °C, which has been reported in the previous works (T. Ghoshal, et. al, Journal of Materials Chemistry 2012, 22, 22949). Furthermore, the presence of metal Pd further stabilize the Ce_2O_3 . We have image to demonstrate that the catalyst is stable after air calcination at 200 °C as shown below.

Supplementary Figure for the reviewers. HAADF-STEM (a, b) images and STEM-EDS mapping (c, d) of the Pd/CeO₂-AT-S-2000 catalyst. (The treated Pd/CeO₂-AT-S catalyst by oxidizing at 200°C for 30 min in air is denoted as Pd/CeO₂-AT-S-2000)

12. Determining the Ce^{3+} content in the catalyst through XANES fitting may not be appropriated. The calculation of material content using XANES should be done through non-linear least-squares fitting. The XPS measurement may one of the proper characterizations to determine the Ce^{3+} species in CeO_2 .

Response: We thank the reviewer for the valuable suggestions. In the original submission, the content of Ce species was analyzed by XANES spectra based on method reported from the literature (Nachimuthu, P. et al., Journal of Solid State Chemistry 2000, 149, 408. Fernandes, V. et al. Journal of Physics Condensed Matter An Institute of Physics Journal 2010, 22, 216004. Bernardi et al., Physical Chemistry Chemical Physics-Cambridge-Royal Society of Chemistry 2015.). In that analysis, the XANES spectra of all the samples were fitted with

Gaussian functions and an arctan function was also included in the fit to exclude the edge jump from XANES spectra. To address the reviewer's concern, we have further tested and analyzed the Ce content on the surface of the catalyst by XPS. According to the XPS results, the content of Ce^{3+} on the surface of Pd/CeO₂-AT-S was significantly higher than that of Pd/CeO₂-AT-S-O, which was consistent with the XANES results.

Revisions:

In the Line 14 of Page 10 of the revised manuscript, we have added “while the proportion of Ce^{3+} was reduced (Supplementary Fig. 13, 14 and Supplementary Table 4).”

In the revised Supplementary Information, we have added the following Figure below as Supplementary Fig. 14 to show the XPS spectra of Ce 3d for the Pd/CeO₂-AT-S-O and Pd/CeO₂-AT-S catalysts.”

Supplementary Figure for the reviewers. XPS spectra of Ce 3d for the Pd/CeO₂-AT-S-O and Pd/CeO₂-AT-S catalysts. It is listed as the Supplementary Fig. 14 in the revised manuscript.

REVIEWERS' COMMENTS

Reviewer #1 (Remarks to the Author):

I am satisfied with the changes made by the authors in the revised manuscript and recommend it for publication as it is.

Reviewer #3 (Remarks to the Author):

The authors have addressed all of my concerns properly, and I believe the paper should be published in its current form in Nature Communications.

Reviewer #4 (Remarks to the Author):

The manuscript has been thoroughly and satisfactorily revised. I recommend this manuscript for publication in Nature Communications.